# CAP2 is a regulator of actin pointed end dynamics and myofibrillogenesis in cardiac muscle

Mert Colpan[1], Jessika Iwanski [1] & Carol C. Gregorio [1✉]

The precise assembly of actin-based thin filaments is crucial for muscle contraction. Dysregulation of actin dynamics at thin filament pointed ends results in skeletal and cardiac myopathies. Here, we discovered adenylyl cyclase-associated protein 2 (CAP2) as a unique component of thin filament pointed ends in cardiac muscle. CAP2 has critical functions in cardiomyocytes as it depolymerizes and inhibits actin incorporation into thin filaments. Strikingly distinct from other pointed-end proteins, CAP2's function is not enhanced but inhibited by tropomyosin and it does not directly control thin filament lengths. Furthermore, CAP2 plays an essential role in cardiomyocyte maturation by modulating pre-sarcomeric actin assembly and regulating α-actin composition in mature thin filaments. Identification of CAP2's multifunctional roles provides missing links in our understanding of how thin filament architecture is regulated in striated muscle and it reveals there are additional factors, beyond Tmod1 and Lmod2, that modulate actin dynamics at thin filament pointed ends.

[1] Department of Cellular and Molecular Medicine and Sarver Molecular Cardiovascular Research Program, The University of Arizona, Tucson, AZ, USA.
✉email: gregorio@email.arizona.edu

Dynamic actin filament assembly and turnover underlie a plethora of biological processes such as cellular morphogenesis and force generation. One of the most striking examples of tightly regulated actin filament assembly can be found in striated muscle. Actin-based thin filaments in striated muscle cells assemble with remarkable precision, which is required for efficient interaction with myosin-based thick filaments, providing the fundamental basis for contractile activity. The development and maintenance of striated muscle depends on the sequential exchange of three α-actin isoforms: α-cardiac, α-skeletal, and α-smooth muscle actin that integrate into thin filaments via polymerization in a highly precise fashion (see ref. [1] for review). Although the mechanisms underlying regulation of α-actin exchange and thin filament assembly are largely unknown, it is known that regulation of filament lengths is accomplished by modulation of actin polymerization at the pointed end of thin filaments in striated muscle[2]. Alterations in thin filament lengths (TFLs) are linked to the development of human muscular diseases including dilated cardiomyopathy (DCM) and nemaline myopathy[3–6]. Members of the tropomodulin family are the only proteins known to localize and exclusively function at thin filament pointed ends in mammalian muscle, with tropomodulin 1 (Tmod1) and leiomodin 2 (Lmod2) as the major isoforms expressed in cardiac tissue[7–10]. In vitro, Tmod1 is a "capping" protein that inhibits actin polymerization and depolymerization from the pointed end[11], while Lmod2 enhances polymerization by its actin-nucleating ability[9]. In myocytes, these proteins coordinate and fine-tune TFLs, where Tmod1 prevents, while Lmod2 promotes elongation of thin filaments (see refs. [12,13] for reviews). Both proteins are indispensable for heart development and function: Tmod1 knockout (KO) mice are embryonic lethal due to cardiac defects[14] and Lmod2-KO mice die ~3 weeks after birth with short thin filaments and severe DCM[15].

Recent evidence shows that adenylyl cyclase-associated protein 2 (CAP2), an actin-binding protein, localizes to the center of the sarcomere (M-line), potentially in close proximity to thin filament pointed ends in the heart[16]. Deletion of Cap2 in mice results in conduction abnormalities and DCM, leading to sudden cardiac death[17,18], a delay in myofibril differentiation, and motor function deficits in neonatal skeletal muscle[19]. Interestingly, deletions in chromosome 6 (which includes the CAP2 gene) in human patients with 6p22 syndrome cause congenital anomalies including heart defects[20–22]. Furthermore, a deleterious mutation in CAP2 was identified in human patients, leading to the development of severe DCM and fatal congestive heart failure[23]. Although all data to date suggest that CAP2 is essential for a viable heart, its in vivo function and exact role in the sarcomere, particularly in thin filament regulation, is largely unknown.

CAP1 and CAP2 are mammalian isoforms of a single ancestral protein, CAP, which was first identified in Saccharomyces cerevisiae as an adenylyl cyclase-binding protein[24]. CAP is important for normal actin organization in a wide-range of organisms including Caenorhabditis elegans[25] and zebrafish[26]. CAP accelerates F-actin disassembly in the presence of actin-depolymerization factor (ADF)/cofilin[27,28] and through its interaction with profilin, catalyzes nucleotide exchange of ADP-actin monomers, recharging them for another round of polymerization[27,29]. The N-terminal region of CAP consists of a coiled-coil and a helical-folded domain (HFD)[30,31], whereas the C-terminal region contains two poly-proline-rich domains, a Wiskott–Aldrich-homology 2 (WH2) domain, and a CAP-retinitis pigmentosa (CARP) domain that is comprised of ß-sheets[32,33]. The coiled-coil region of human CAP1 and CAP2 allows tetramers of HFDs to form, which increases cofilin-dependent actin depolymerization[34]. Mammalian CAPs interact with filamentous actin (F-actin) via the HFD[31,34,35] and with globular actin (G-actin) using the WH2 and CARP domains[33].

In this manuscript, we deciphered that CAP2 is a unique component of the thin filament pointed end protein complex, whose primary function is to regulate thin filament architecture. Specifically, we discovered that CAP2: (1) is essential for myocyte maturation; (2) depolymerizes actin filaments in a tropomyosin (Tpm)-dependent manner; and (3) inhibits actin incorporation and alters actin dynamics within thin filaments. We found that the mechanism of CAP's pointed end function is to regulate α-actin isoform exchange by promoting replacement of α-smooth and α-skeletal with α-cardiac actin. Our results reveal that CAP2 is critical in the transition of nascent pre-myofibrils into mature myofilaments, as well as in facilitating proper differentiation and sarcomere remodeling in cardiomyocytes.

## Results

**CAP2 localizes to thin filament pointed ends**. CAP2 was shown to localize to the M-line of sarcomeres in mouse myofibrils, based on co-staining for an M-line protein, myomesin[16]. However, when myofibrils are contracted, thin filament pointed ends and M-lines of the sarcomere overlap (they are not distinguishable from each other by immunofluorescence microscopy). Therefore, it is not known whether CAP2 assembles at the M-line or pointed ends.

To determine the precise localization of CAP2, we investigated its subcellular assembly in embryonic chick cardiomyocytes, in line with similar experiments previously conducted to determine the localization of Tmod1 and Lmod2[10,36]. Immunostaining for endogenous CAP2 and GFP-CAP2 revealed the same pattern of assembly, where CAP2 striations were observed near the M-lines of sarcomeres (across the Z-discs), as well as diffusely distributed in the cytoplasm (Fig. 1a, b). CAP2 striations could be resolved into doublets (Fig. 1a, b), indicative of localization to the adjacent pointed ends of thin filaments. To validate this observation, we performed structured illumination super-resolution microscopy (SR-SIM) on cardiomyocytes expressing GFP-CAP2 and co-stained for Tmod1 or myomesin to mark thin filament pointed ends and M-lines, respectively. When myofibrils were contracted, CAP2 staining appeared as a single line near the M-line, but when myofibrils were relaxed, the staining indeed appeared as doublets, corresponding to thin filament pointed ends (Fig. 1c, d). Consistent with pointed end assembly, doublets of GFP-CAP2 flank myomesin and co-localize with Tmod1. These results strongly suggest that CAP2 is an actin filament pointed end-binding protein.

**CAP2 has a distinct temporal expression compared with Tmod1 and Lmod2**. To understand the role of CAP2 in myofibrillogenesis, we examined the expression levels of CAP2 in neonatal rat cardiomyocytes (NRCMs). Interestingly, CAP2 is expressed at higher levels earlier in culture with its levels decreasing as cardiomyocytes matured (Fig. 2a, b). Conversely, Lmod2 levels increased throughout maturation, consistent with previous studies[36], while Tmod1 levels remained constant (Fig. 2c, d). Therefore, CAP2's expression pattern differs from Tmod1 and Lmod2, suggesting that it has a unique role(s) at thin filament pointed ends during early myofibrillogenesis.

**CAP2 associates with a dynamic population of actin filaments**. In order to decipher CAP2's role in myofibrillogenesis, we performed numerous assays to determine its mechanism of action. We first addressed whether disrupting actin equilibrium in cardiomyocytes would affect the subcellular assembly of CAP2. We treated NRCMs with latrunculin A (Lat A), which sequesters G-

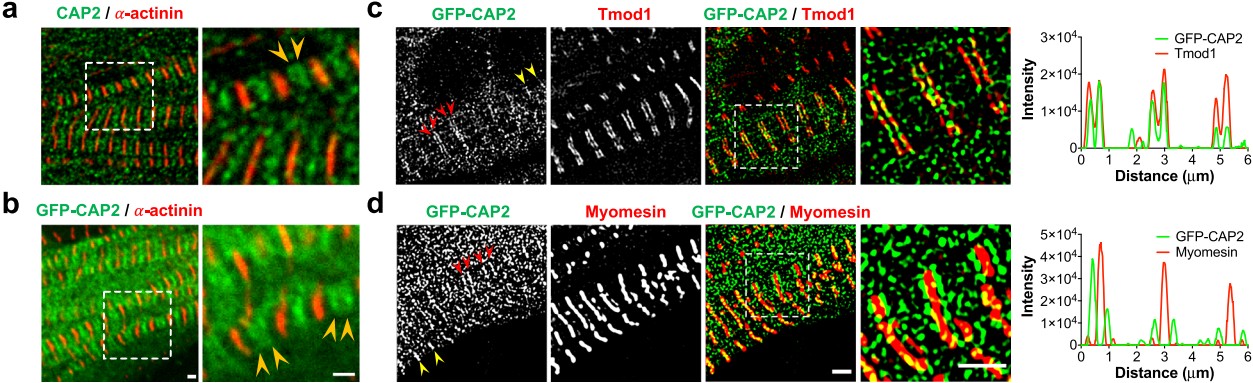

**Fig. 1 CAP2 localizes to the pointed ends of thin filaments.** Subcellular assembly of CAP2 in chick embryonic cardiomyocytes was investigated by **a** immunostaining for endogenous CAP2 and by **b** expressing GFP-CAP2. Anti-α-actinin (red) antibodies were used to mark the Z-discs of the sarcomeres. GFP-CAP2-expressing cells were stained for **c** Tmod1, or **d** myomesin (red) to mark thin filament pointed ends or M-lines, respectively, and subjected to super resolution-structured illumination microscopy (SR-SIM). The intensity profiles of stained proteins reveal that GFP-CAP2 resolves both into singlets (yellow arrowheads) and into doublets at the pointed ends (red arrowheads) based on the contractile state. The line intensity profile of GFP-CAP2 doublets (green lines) colocalize with the profile of Tmod1, but not with that of myomesin (red lines). Scale bar = 1 μm.

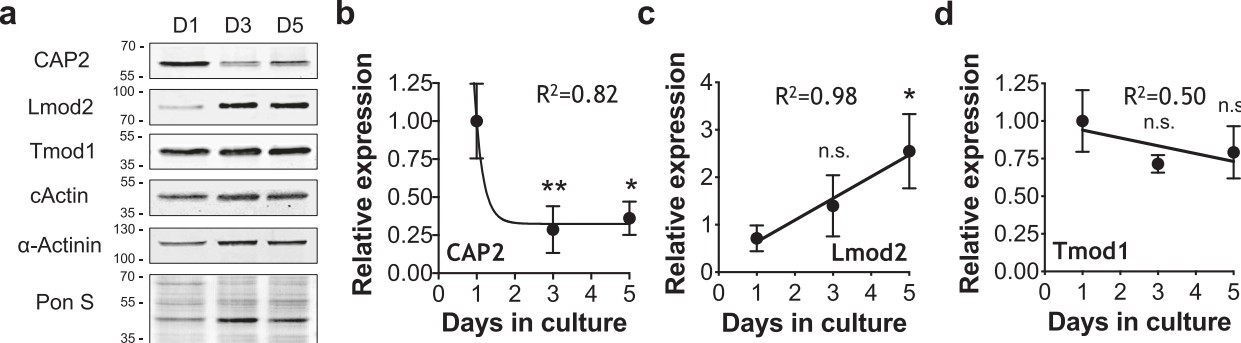

**Fig. 2 The levels of CAP2 protein decrease in neonatal rat cardiomyocytes during myofibrillogenesis in culture. a** Immunoblot analysis of proteins from whole cell lysates on days 1, 3, and 5 after plating. Quantification of immunoblots show compared to day 1, **b** decreasing CAP2 levels (**$p = 0.0066$, *$p = 0.0112$, $F = 14.37$), **c** increasing Lmod2 levels (n.s.: $p = 0.4078$, *$p = 0.0236$, $F = 6.985$) and **d** unchanged Tmod1 levels (n.s.: $p = 0.1499$, n.s.: $p = 0.3140$, $F = 2.588$) as cardiomyocytes mature (mean ± SD, $n = 3$ cultures, one-way ANOVA, df = 2, n.s.: not significant).

actin and perturbs actin polymerization. Note, Lat A treatment does not disassemble mature myofibrils at the concentrations used here (potentially due to stabilization by capping proteins and/or Tpm[37]). Instead, latrunculins are known to disrupt a more dynamic (less stable) population of actin filaments in cardiomyocytes[38]. Following Lat A treatment, CAP2 lost its striated distribution and became more diffuse in NRCMs (Fig. 3a, b). This result suggests that assembly of CAP2 at the pointed end is dependent on active actin polymerization in cardiomyocytes. Cellular localization of Tmod1 was not altered in the presence of Lat A (Fig. 3c, d) [similar to results reported for Tmod1 upon Latrunculin B treatment[36]]. These results reveal that Tmod1 associates with more mature and stabilized filaments, whereas CAP2 associates with a dynamic population of parallel filaments that are potentially not stabilized by Tmod1 and/or Tpm. Alternatively, an increase in G-actin by Lat A treatment recruits CAP2, but not Tmod1, away from pointed ends.

**Excess levels of CAP2 has little effect on TFLs.** We next aimed to determine if CAP2 has an effect on TFL regulation. It is known that altered expression levels of the pointed end proteins Lmod2 and Tmod1, significantly changes TFLs in cardiomyocytes [for reviews see refs. [12,13]]. We hypothesized that altering CAP2 levels would also affect TFLs. To test this hypothesis, TFLs were measured in NRCMs expressing GFP-CAP2, with a ~15-fold excess

over endogenous CAP2 levels (Fig. S1a, b). Surprisingly, excess CAP2 led to a small, yet significant decrease (~2% lower than control cells) in TFLs (Fig. S1c–f). Note, increased levels of Tmod1 or Lmod2 lead to dramatic and physiologically relevant changes in TFLs [i.e., ~10% longer in hearts of Lmod2-transgenic mice[39] and ~10% shorter GFP-Tmod1-expressing cardiomyocytes[2]]. Therefore, CAP2 does not function similar to Tmod1 or Lmod2, and it has other important role(s) at the pointed end of thin filaments.

**CAP2 depolymerizes actin filaments.** To decipher CAP2's specific function at thin filament pointed ends, we tested the effect of CAP2 on the stabilization of actin filaments. NRCMs expressing GFP or GFP-CAP2 were lysed to fractionate G-actin and F-actin by ultracentrifugation. Quantification of immunoblots revealed that cardiomyocytes expressing GFP-CAP2 resulted in a lower percentage of F-actin (49.5 ± 3.4%) compared to GFP-expressing cardiomyocytes (60.6 ± 3.7%) (Fig. 4a, b). To confirm the validity of our assay, we performed the same experiment on cells treated with Lat A and jasplakinolide, which depolymerizes and stabilizes actin filaments, respectively. As expected, F-actin levels (71.5 ± 7.0%) decreased in cells treated with Lat A (18.5 ± 5.6%) but increased in cells treated with jasplakinolide (92.3 ± 3.4%) (Fig. 4c, d). Collectively, this result is the first to hint at CAP2's distinct role in cardiac muscle: it promotes depolymerization of

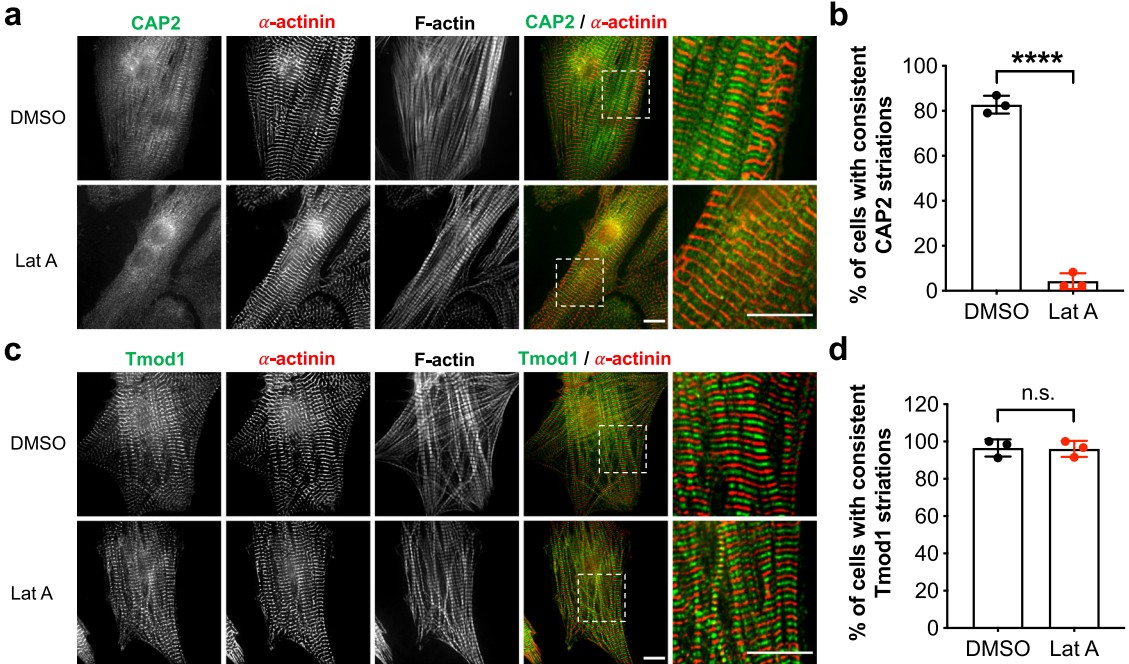

**Fig. 3 Depletion of actin monomers disrupts the assembly of CAP2, but not Tmod1 in cardiomyocytes.** Neonatal rat cardiomyocytes were treated with dimethyl sulfoxide (DMSO, control) or latrunculin A (Lat A) and fixed. Immunostaining was done using: **a** anti-CAP2 (green), or **b** anti-Tmod1 (green) and anti-α-actinin (red) antibodies, and phalloidin to probe for F-actin. Scale bar = 10 μm. **c, d** % of cells showing consistent CAP2 (black dots) or Tmod1 (red dots) pointed end striations, respectively (mean ± SD, 30 cells/culture, $n$ = 3 cultures, ****$p$ < 0.0001, $t$ = 25.71; n.s.: $p$ = 0.8975, $t$ = 0.1373, Student's $t$-test, df = 4, n.s.: not significant).

F-actin into G-actin and it is a new regulator of actin filament disassembly in cardiomyocytes.

**CAP2 and Tpm do not interact.** Tpm is a major component of thin filaments in striated muscle and the respective functions of Tmod1 and Lmod2 are enhanced by interacting with Tpm[9,11,40]. To determine whether CAP2's function in actin depolymerization is also regulated by interacting with Tpm, we performed affinity pull-down experiments with GFP or GFP-tagged CAP2, Tmod1, or Lmod2 expressed in NRCMs (Fig. S2a). Pull-down of endogenous Tpm was detected with GFP-Tmod1 and GFP-Lmod2, but not with GFP-CAP2. Pull-down of endogenous actin was detected with all three proteins. Nondenaturing-PAGE experiments with purified recombinant proteins also showed an interaction between Tmod1 and Tpm1.1 or actin, but CAP2 only interacted with actin (Fig. S2b). Therefore, in two independent assays an interaction between CAP2 and Tpm1.1 did not occur.

**Tpm inhibits the depolymerization activity of CAP2.** Biochemically, CAP2 accelerates ADF/cofilin-dependent disassembly of F-actin[27,34] but this function has not been assessed with Tpm-containing thin filaments, as in the case of muscle cells. Since CAP2 and Tpm1.1 do not appear to interact, we investigated if Tpm has an indirect effect on CAP2's thin filament disassembly function. F-actin with or without Tpm1.1 was depolymerized in the presence of a muscle-specific cofilin isoform (cofilin-2; CFL2) and CAP2. Ultracentrifugation separated G-actin to the supernatant and F-actin to the pellet. Actin band density in the pellets and supernatants was quantified to calculate the amount of depolymerized actin (Fig. 4e). In the absence of Tpm1.1, CAP2 accelerated the depolymerization of CFL2-decorated filaments (Fig. 4f). However, F-actin was significantly resistant to disassembly by CAP2 in the presence of Tpm. At all concentrations of CAP2 tested, filament disassembly was stronger than with CFL2 only. Furthermore, the presence of Tpm inhibited CAP2's

depolymerization activity at all CAP2 concentrations tested. Hence, Tpm inhibits the depolymerization activity of CAP2.

To further validate Tpm's ability to regulate the depolymerization activity of CAP2, NRCMs expressing mCherry or mCherry-Tpm1.1 in combination with GFP or GFP-CAP2 were lysed and the percentages of G-actin and F-actin levels were quantified (Fig. 4g). Expression of excess levels of Tpm1.1 increased the percentage of F-actin levels (69.6 ± 6.8 to 83.2 ± 3.4%) (Fig. 4h) indicative of an increased number of filaments. GFP-CAP2 was ineffective in depolymerizing the additional filaments that were created by mCherry-Tpm1.1 (82.4 ± 4.2%). Thus, saturation by Tpm stabilizes actin filaments and inhibits their disassembly by CAP2.

**Tpm1.1 recruits Tmod1, not CAP2 to thin filaments.** Next, we investigated whether mCherry-Tpm1.1-stabilized filaments could recruit endogenous CAP2. Immunoblot analysis demonstrated that the percent distribution of CAP2 in the F-actin populations of cardiomyocytes expressing mCherry (28.6 ± 5.9%) or mCherry-Tpm1.1 (25.0 ± 9.6%) was not different (Fig. 4i, j). However, the percentage of Tmod1 in the F-actin fraction was increased (42.7 ± 2.6% to 65.6 ± 3.0%) with excess Tpm1.1 (Fig. 4k). Therefore, the additional filaments that are stabilized by excess levels of Tpm1.1 (Fig. 4h) recruit Tmod1 (due to its known Tpm-binding sites and thus increased affinity) but not CAP2, to their pointed ends. This suggests that the binding of CAP2 to actin filaments is ineffective when stabilized by Tpm1.1.

**CAP2 inhibits incorporation of actin onto thin filament pointed ends.** We next aimed to determine if CAP2 has an effect on actin incorporation into thin filaments. To determine how CAP2 affects the kinetics of actin polymerization, fluorescence recovery after photobleaching (FRAP) was performed in NRCMs expressing GFP-cActin and either mCherry alone or mCherry-CAP2. GFP-cActin was photobleached within multiple

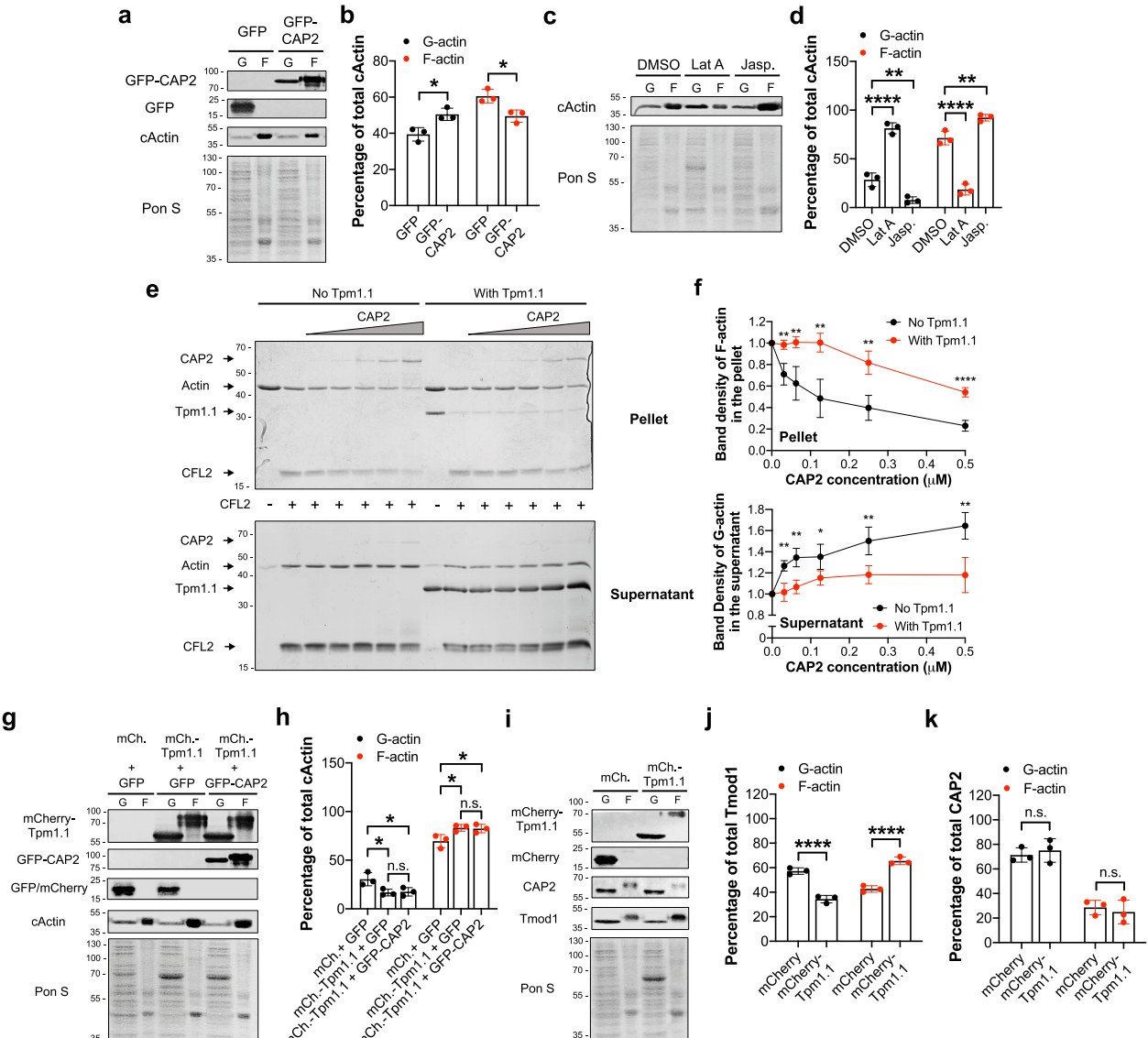

**Fig. 4 CAP2 depolymerizes actin filaments but stabilization by tropomyosin prevents CAP2-mediated filament disassembly. a** Immunoblot analysis for GFP, globular (G-actin, G) and filamentous (F-actin, F) cardiac actin (cActin) in neonatal rat cardiomyocytes (NRCMs) transduced with GFP or GFP-CAP2 adenovirus. **b** Quantification of immunoblots demonstrates that the percentage of cardiac F-actin (red dots) was significantly lower in GFP-CAP2-expressing cells (49.5 ± 3.4%) compared to GFP-expressing control cells (60.6 ± 3.7%) (mean ± SD, $n = 3$ cultures, $*p = 0.0105$, two-way ANOVA $F = 28.78$, df = 1). **c** Immunoblot analysis for G-actin or F-actin from NRCMs treated with 0.01% dimethyl sulfoxide (DMSO), 2 μM Latrunculin A (Lat A), or 0.15 μM jasplakinolide (Jasp.) overnight. **d** Quantification of immunoblots demonstrates that Lat A treatment results in less (71.5 ± 7.0 to 18.5 ± 5.6%), whereas jasplakinolide (Jasp.) treatment leads to more F-actin (92.3 ± 3.4%, red dots), respectively (mean ± SD, $n = 3$ cultures, $****p < 0.0001$, $**p = 0.0038$, two-way ANOVA $F = 280.3$, df = 2). **e** Actin filaments (5 μM) in the absence or presence of 5 μM striated muscle α-tropomyosin (Tpm1.1) were diluted five-fold and depolymerized with 2 μM cofilin-2 (CFL2) and increasing concentrations of CAP2 (0, 0.03, 0.06, 0.12, 0.25, 0.5 μM). The mixtures were ultracentrifuged, and the supernatants and pellets were subsequently separated by SDS–PAGE. **f** Band densities (mean ± SD) of actin relative to the control (actin in the presence of CFL2) were quantified to calculate the fraction of actin in the pellets ($n = 4$, $**p = 0.0023$, $t = 5.056$; $**p = 0.0035$, $t = 4.639$; $**p = 0.0020$, $t = 5.222$; $**p = 0.0019$, $t = 5.286$; $****p < 0.0001$, $t = 9.326$, Student's $t$-test, df = 6) and the supernatants ($n = 4$, $**p = 0.0024$, $t = 5.038$; $**p = 0.0021$, $t = 5.140$; $*p = 0.0267$, $t = 2.919$; $**p = 0.0064$, $t = 4.099$; $**p = 0.0044$, $t = 4.445$, Student's $t$-test, df = 6) upon depolymerization in the absence (black dots) or presence of Tpm1.1 (red dots). **g** Immunoblots for mCherry (mCh.), GFP, G-actin and F-actin in NRCMs transduced with GFP, mCherry, mCherry-Tpm1.1, and/or GFP-CAP2 adenovirus. **h** Quantification of immunoblots demonstrates the percentage of F-actin (red dots) was significantly higher in mCherry-Tpm1.1-expressing cells (83.2 ± 3.4%) compared to mCherry-expressing control cells (69.6 ± 6.8%). Furthermore, GFP-CAP2 was ineffective in depolymerizing actin when co-expressed with mCherry-Tpm1.1 (82.4 ± 4.2%) (mean ± SD, $n = 3$ cultures, $*p = 0.0347$, $*p = 0.0489$, n.s.: $p > 0.9999$, two-way ANOVA, $F = 14.10$, df = 2). **i** Immunoblot analysis for G-actin and F-actin, CAP2 and Tmod1 in NRCMs transduced with mCherry or mCherry-Tpm1.1. Quantification of immunoblots demonstrate that **j** the fraction of CAP2 associated with F-actin (28.6 ± 5.9%) was unaffected by mCherry-Tpm1.1 expression (25.0 ± 9.6%, red dots), whereas **k** higher levels of Tmod1 were recruited to the F-actin fraction in mCherry-Tpm1.1-expressing cells (65.6 ± 3.0%) compared to mCherry-expressing cells (42.7 ± 2.6%, red dots) (mean ± SD, $n = 3$ cultures, n.s.: $p = 0.8289$, $F = 0.6425$; $****p < 0.0001$, $F = 196.3$, two-way ANOVA, df = 1, n.s.: not significant).

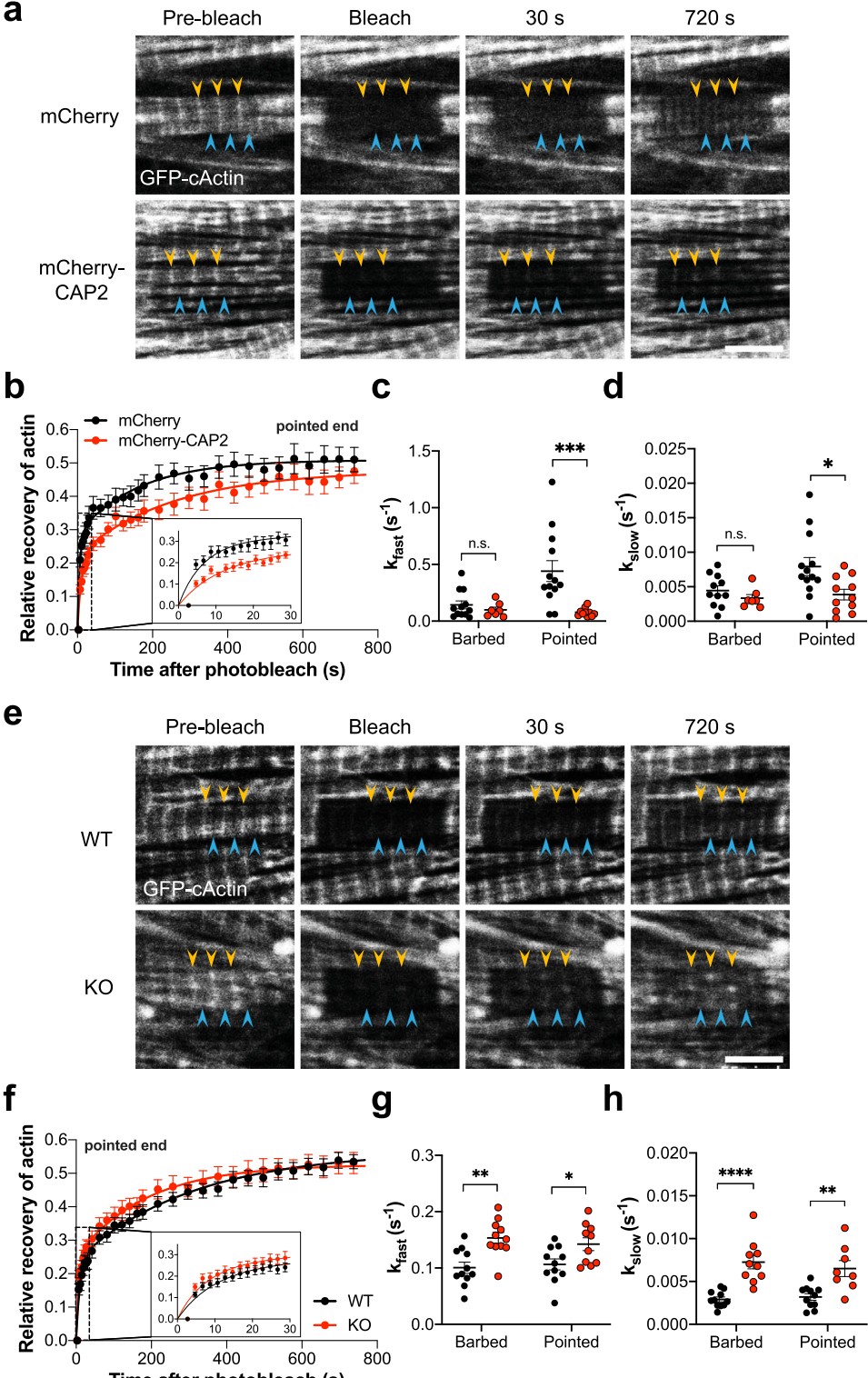

sarcomeres and fluorescence recovery was followed independently at the pointed and barbed ends of thin filaments (Fig. 5a). Actin incorporation at pointed ends was significantly slower (smaller rate constant, larger halftime) predominantly in the fast phase of recovery in cells transduced with mCherry-CAP2 compared with mCherry (Fig. 5b–d, Table S1). Although, the rate of actin incorporation at the pointed ends was diminished by CAP2, the mobile fraction at the slow phase of recovery was unaffected, indicating that the amount of actin assembled at the end of the experiment was unchanged between cells transduced with mCherry-CAP2 and mCherry. None of the parameters used to measure actin incorporation at the barbed ends of filaments were altered in the presence of mCherry-CAP2 (Table S1, Fig. S3a). Therefore, CAP2 produces a less dynamic filament by slowing down (inhibiting) actin incorporation, specifically at the pointed end of thin filaments.

**Fig. 5 CAP2 inhibits actin dynamics by delaying actin incorporation to thin filaments in cardiomyocytes. a** Representative images of fluorescence recovery after photobleaching (FRAP) of GFP-cardiac actin (cActin) in neonatal rat cardiomyocytes transduced with mCherry or mCherry-CAP2. Barbed (yellow arrowheads) and pointed (blue arrowheads) ends of actin filaments are marked. **b** Mean relative recovery at thin filament pointed ends after photobleaching over time in mCherry (black dots) or mCherry-CAP2-expressing cells (red dots). Rate constants ($k$) for the **c** fast and **d** slow phases of association of GFP-cActin in mCherry (black dots) or mCherry-CAP2-expressing cells (red dots) are indicated as mean ± SEM ($n = 11, 7, 13, 10$ cells from three independent cultures; n.s.: $p = 0.9671$, ***$p = 0.0006$, $F = 6.330$; n.s.: $p = 0.8908$, *$p = 0.0204$, $F = 19.10$, two-way ANOVA, df = 1, n.s.: not significant). **e** Representative images of FRAP of GFP-cActin in neonatal WT or *Cap2*-KO mouse cardiomyocytes. Barbed (yellow arrow) and pointed (blue arrowheads) ends of actin filaments are marked. **f** Mean relative recovery at thin filament pointed ends after photobleaching over time in WT (black dots) or *Cap2*-KO cardiomyocytes (red dots), fit using nonlinear regression curves with a two-phase-exponential association equation. Rate constants ($k$) for the **g** fast and **h** slow phases of association of GFP-cActin in WT (black dots) or *Cap2*-KO cardiomyocytes (red dots) are indicated as mean ± SEM ($n = 11, 10, 11, 8$ cells from three independent cultures; **$p = 0.0012$, *$p = 0.0356$, $F = 19.10$; ****$p < 0.0001$, **$p = 0.0012$, $F = 40.31$, two-way ANOVA, df = 1). Scale bar = 5 μm.

**Actin incorporation into thin filaments is faster in *Cap2*-KO cardiomyocytes.** In order to corroborate our finding that CAP2 inhibits actin incorporation and to gain more mechanistic insight into the functional properties of CAP2, we tested the kinetics of actin assembly in neonatal cardiomyocytes from *Cap2*-KO mice. FRAP was performed in isolated cardiomyocytes expressing GFP-cActin. Fluorescence recovery of actin was followed independently at the pointed and barbed ends of thin filaments (Fig. 5e). The mean rate constant was larger and halftime was smaller for both fast and slow phases of recovery in *Cap2*-KO cells compared to WT cells (Fig. 5f–h and Table S2). Interestingly, actin recovery in *Cap2*-KO cardiomyocytes was more dynamic at both barbed and pointed ends of thin filaments compared to WT cardiomyocytes (Figs. 5f–h and S3b). In alignment with FRAP results obtained from cardiomyocytes overexpressing CAP2, the mobile fraction of actin at the end of recovery (slow phase) was unchanged by the absence of CAP2. Therefore, loss of CAP2 does not change the total amount of actin incorporated into filaments but instead it accelerates its incorporation and increases the dynamics of its polymerization.

**Cap2-null cardiomyocytes display a delay in maturation and actin disorganization.** Thus far we have demonstrated that (1) CAP2 plays an important role in the regulation of actin dynamics at the pointed end of thin filaments and (2) it has very distinct functions from Lmod2 and Tmod1. However, due to the specific temporal expression of these proteins (Fig. 2), the role of CAP2 in myofibrillogenesis remains to be elucidated. Therefore, to determine how cellular features of cardiac muscle are affected by the absence of CAP2, we examined isolated WT and *Cap2*-KO cardiomyocytes to visualize potential differences linked to myofibrillogenesis between them. *Cap2*-KO cardiomyocytes appeared rounder, less-rod shaped and smaller in size compared to WT cells, most strikingly at earlier stages of myofibrillogenesis (D2), suggesting compromised cell spreading and maturity (Fig. S4).

Next, we investigated the subcellular structure of WT and *Cap2*-KO cardiomyocytes via deconvolution immunofluorescence microscopy. Phalloidin staining of *Cap2*-KO cells revealed disorganized actin filaments throughout the sarcoplasm without an ordered array of sarcomeric striations (Fig. 6a). Unlike WT cardiomyocytes, α-actinin staining in *Cap2*-KO cells displayed a punctate pattern, reminiscent of nascent, immature pre-myofibrils[41]. By day 5 in culture, increased sarcomere alignment was observed in *Cap2*-KO cells; however, the myofibrils were still more disorganized (less mature) than in WT cells (Fig. 6b). These results demonstrate that the transition from nascent myofibrils to mature myofilaments is delayed in *Cap2*-KO cells. Interestingly, no noticeable alterations in the assemblies of thick filament proteins (myosin-binding protein C and myomesin) were observed, (Fig. S5a, b). Therefore, the delay in myofibrillogenesis

within *Cap2*-KO cardiomyocytes likely originates from the absence of CAP2's role in modulating actin filament architecture. TFLs and Tmod1 localization in *Cap2*-null cardiomyocytes were comparable to WT cardiomyocytes (Fig. S6), further highlighting that CAP2 has little effect on determining TFLs.

**Cap2-null cardiomyocytes display altered α-actin assembly and expression.** Since *Cap2*-KO cells demonstrated delayed myofibrillogenesis, we investigated subcellular assemblies of known myofibril maturation markers in cardiomyocytes: specifically, α-SMA, desmin and nonmuscle myosin IIB, which are components of thin filaments, intermediate filaments and plasma membrane/pre-myofibrils, respectively[42–45]. Immunofluorescence microscopy demonstrated no notable difference in desmin and nonmuscle myosin IIB staining between WT and KO cardiomyocytes (Fig. S5b, c). However, the fluorescence intensity of α-SMA staining was remarkably higher and α-SMA was incorporated into filaments in *Cap2*-KO cells compared to WT cells (Fig. 6c).

The striking difference in α-SMA incorporation in *Cap2*-KO cardiomyocytes prompted us to investigate two other α-actin isoforms: α-cardiac actin (cActin) and α-skeletal muscle actin (α-SKA). Although cActin assembled into thin filaments in WT and *Cap2*-KO cells, it demonstrated an irregular alignment with the phalloidin stain, such that cActin striations were disorganized in *Cap2*-KO cardiomyocytes (Fig. 6d). α-SKA showed weak to no assembly in WT cells (Fig. 6e). In contrast, there was a clear striated pattern in *Cap2*-KO cells for α-SKA localized to the pointed ends of thin filaments (Fig. 6e).

Immunoblots revealed that while cActin levels were unchanged, α-SMA and α-SKA were significantly upregulated in *Cap2*-KO cardiomyocytes (Fig. 6f). Additionally, *Cap2*-KO cardiomyocytes demonstrated increased G-actin and F-actin levels for both α-SMA and α-SKA but similar levels for cActin (Fig. S7). Collectively, our results indicate that expression levels and assembly of α-SMA and α-SKA are significantly altered in *Cap2*-KO cardiomyocytes, indicating CAP2 plays an important role in the distribution of α-actin isoforms between G-actin and F-actin during cardiac muscle development.

**α-SMA and α-SKA replace cActin in thin filaments from *Cap2*-KO mice.** To test if our observations in cultured *Cap2*-KO cells were physiologically relevant in vivo, we investigated *Cap2*-KO mice. In agreement with previous studies[17,18], male *Cap2*-KO mice demonstrated a reduced survival rate (~50%) due to sudden cardiac death compared to WT mice (Fig. S8a). We also observed reduced body weight and dilation of the heart as early as postnatal day (PD) 15 and reduced left ventricular stroke volume in male *Cap2*-KO mice (Fig. S8b–d). Since these parameters confirmed the presence of pathological disease in our mouse model, we

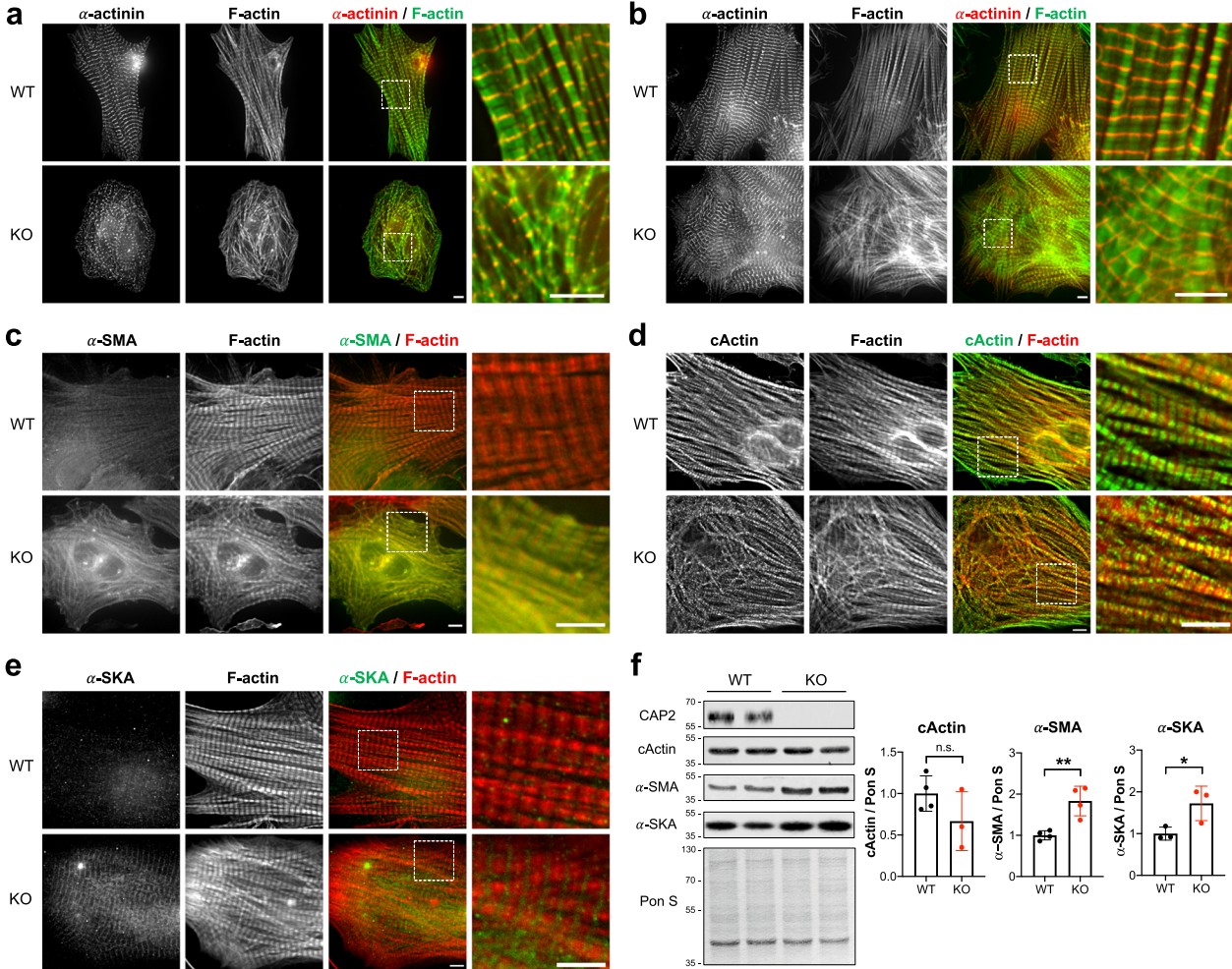

**Fig. 6 *Cap2*-KO cells exhibit thin filament disorganization with higher levels of α-smooth muscle and α-skeletal muscle actin and a delay in maturation.** Cardiomyocytes were isolated from hearts of neonatal WT or *Cap2*-KO mice and fixed **a** 1 day or **b** 5 days after plating for immunofluorescence analysis. Immunostaining of WT or *Cap2*-KO cells was performed using anti-α-actinin (red) antibody, and phalloidin (green) to probe for F-actin. Immunostaining of WT or *Cap2*-KO cardiomyocytes 5 days after plating with **c** anti-α-smooth muscle actin (α-SMA, green) **d** anti-cardiac actin (cActin, green) or **e** anti-skeletal muscle actin (α-SKA, green) antibody, and phalloidin (red) to probe for F-actin. Scale bar = 5 μm. **f** Immunoblot analysis for whole cell lysates of WT (black dots) or *Cap2*-KO cells (red dots) collected 5 days after plating. Quantification of immunoblots shows unchanged levels of cActin but increased levels of α-SMA and α-SKA (~2-fold) in *Cap2*-KO cells (mean ± SD, $n = 3–4$ cultures, n.s.: $p = 0.1780$, $t = 1.567$, df = 5; **$p = 0.0044$, $t = 4.436$, df = 6; *$p = 0.0467$, $t = 2.844$, df = 4, Student's $t$-test, n.s.: not significant).

analyzed the in vivo effects of loss of *Cap2* by first, measuring TFLs from *Cap2*-KO hearts. Similar to our results from cultured *Cap2*-KO cells, TFLs in *Cap2*-KO hearts were unaltered (Fig. S9). Second, immunoblots of *Cap2*-KO hearts on PD 15, 60, and 120 revealed a decrease in cActin protein expression (~30–50%), progressive upregulation of α-SMA protein expression up to ~20-fold by PD 120, and upregulation of α-SKA protein levels by ~70-fold on PD 15 and by ~8-fold on PD 60 and 120 compared to age-matched WT hearts (Fig. 7a–c). The expression of total α-actin was unchanged between the groups. Third, immunostaining of thin filaments from *Cap2*-KO hearts revealed that striations of cActin were shorter (from their pointed ends) compared to WT hearts (Fig. 7d, g). Line scan analysis demonstrated that the length of cActin striations was ~30% shorter in the *Cap2*-KO hearts (Fig. 7e). We did not detect strong assembly of α-SMA and α-SKA in the WT hearts, however *Cap2*-KO hearts showed strong striated assembly for both isoforms (Fig. 7f). Collectively, these findings demonstrate that cActin is substituted by α-SMA and α-SKA in thin filaments from *Cap2*-KO mice, with no overall change in TFLs.

**CAP2 protein expression in mice decreases with age.** Postnatal clearance of α-SMA into cActin and α-SKA is a critical step for α-actin switch and cardiomyocyte maintenance[46–48]. Since we observed impaired turnover of α-SMA and α-SKA into cActin, we investigated whether temporal expression of CAP2 coincides with that of α-actin isoforms. Immunoblots of WT mice on PD 15, 60, and 120 showed that CAP2 protein levels are highest at PD 15 and significantly reduced with age (Fig. 8a, b). We also found distinct expression patterns for α-actin isoforms. α-SMA is expressed at the highest level on PD 15 and decreases with age, while α-SKA is highest on PD 60 (Fig. 8a–d) and cActin is robustly expressed at all ages (Fig. 8a, e). These findings reveal that the expression of α-actin isoforms has a unique sequential pattern in the heart and CAP2 is a factor that participates in their proper clearance and exchange in early postnatal stages.

## Discussion
Actin filament architecture is precisely regulated in the cytoskeleton of most cell types, with the purpose of executing cellular processes such as cell motility and force generation. An important

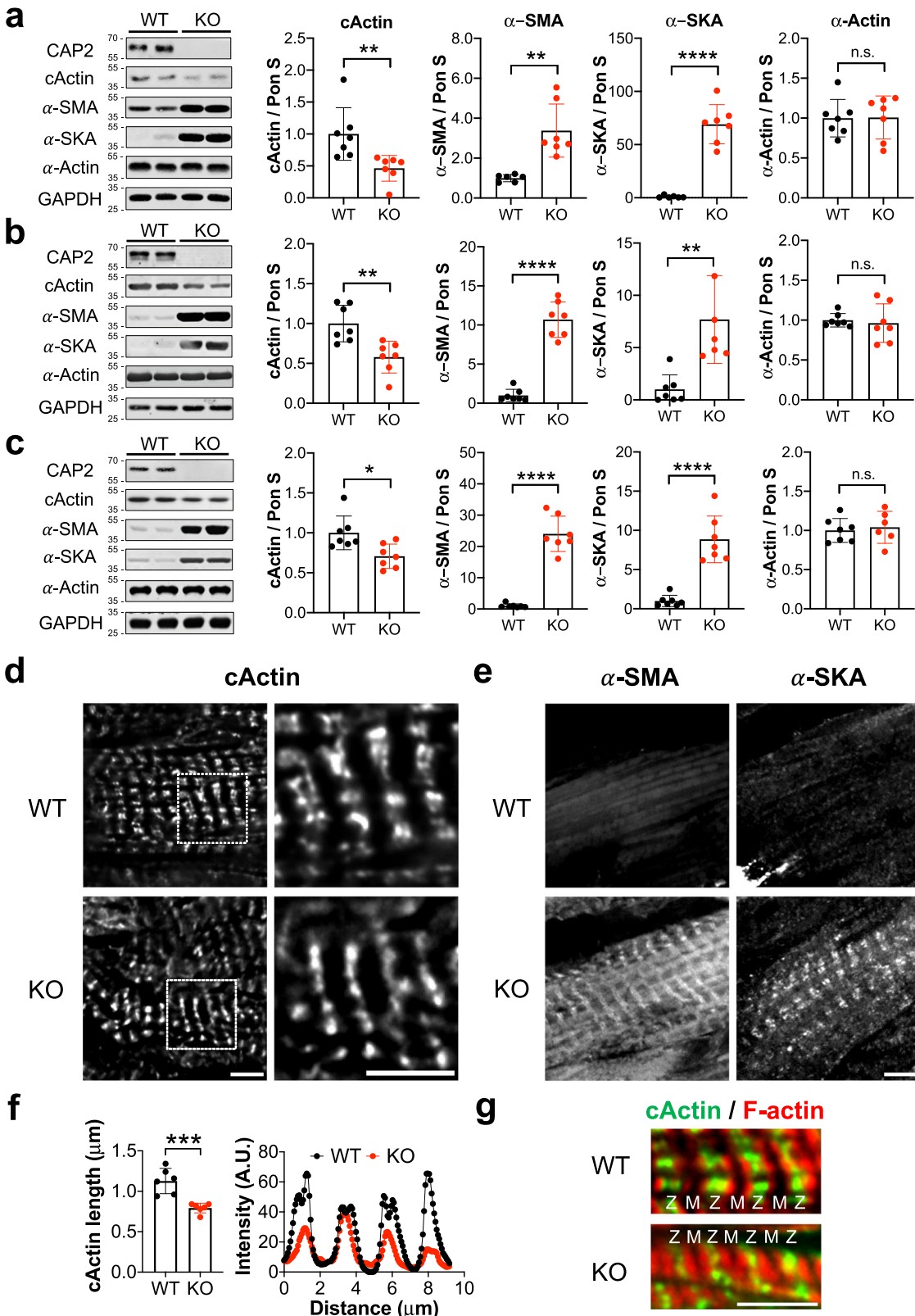

example of thin filament architecture is found in sarcomeres of striated muscle, where their perfectly parallel orientation allows for remarkable resolution of actin filament assembly and structure. Maintenance of thin filament structure is critical for its interaction with thick filaments and for the length–tension relationship during muscle contraction[49]. Regulation of actin filament dynamics, especially at the pointed end, is of great importance, since alterations in TFLs occur at this end of the filament in striated muscle[2] and are linked to life-threatening myopathies in humans[3–6]. Therefore, identifying protein components at pointed ends is important to better understand thin filament regulation and function. Here, we demonstrated that CAP2 is a unique thin filament pointed end-binding protein with new functions in cardiac muscle. We discovered that CAP2

**Fig. 7 Thin filaments from *Cap2*-KO mice are composed of a heterogenous population of actin isoforms.** Immunoblot analysis from left ventricular tissue of WT (black dots) or *Cap2*-KO mice (red dots) at postnatal days (PD) **a** 15 (**p = 0.0092, t = 3.102, df = 12; **p = 0.0012, t = 4.328, df = 11; ****p < 0.0001, t = 8.962, df = 11, n.s.: p = 0.9505, t = 0.06343, df = 12), **b** 60 (**p = 0.0032, t = 3.667, df = 12; ****p < 0.0001, t = 10.64, df = 12, **p = 0.0018, t = 3.993, df = 12, n.s.: p = 0.7179, t = 0.3699, df = 12) and **c** 120 (*p = 0.0116, t = 2.974, df = 12; ****p < 0.0001, t = 10.75, df = 12; ****p < 0.0001, t = 6.788, df = 12; n.s.: p = 0.6945, t = 0.40339, df = 11). Quantification of immunoblots show down-regulation of cardiac actin (cActin), up-regulation of α-smooth muscle actin (α-SMA) and α-skeletal muscle actin (α-SKA) but unchanged levels of total α-actin in *Cap2*-KO mice (mean ± SD, n = 6–7 mice, Student's t-test, n.s.: not significant). Representative immunofluorescence images of PD 120 WT or *Cap2*-KO mice stained with **d** anti-cActin, **e** anti-α-SMA or anti-α-SKA antibodies. **f** Line scan and plot profile analyses reveal shorter lengths for cActin striations in *Cap2*-KO (red dots) mice compared to WT mice (black dots) (mean ± SD, n = 10 measurements from each cell, 6 cells, ***p = 0.0006, t = 4.875, df = 10, Student's t-test). **g** Immunostaining of WT or *Cap2*-KO hearts using anti-cActin (green) and phalloidin for F-actin (red) demonstrate that cActin striations are shortened from the M-lines (M) of *Cap2*-KO thin filaments, as opposed to the Z-disc (Z) end. Scale bar = 5 µm.

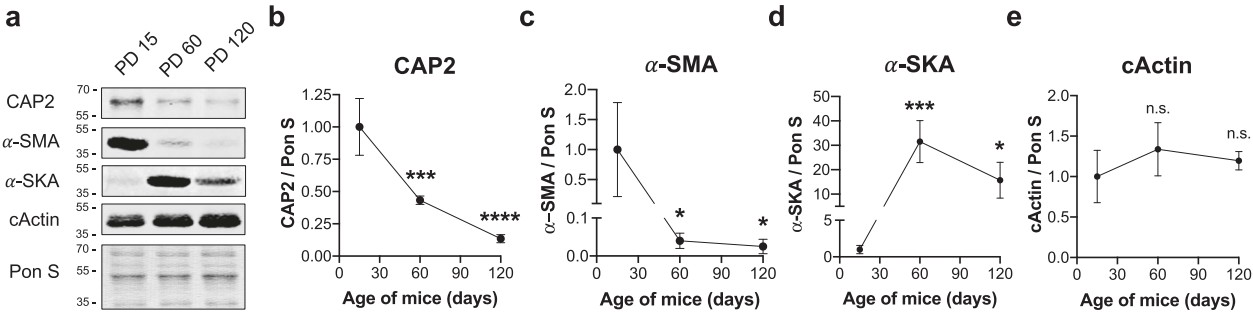

**Fig. 8 The levels of CAP2 protein decrease in mice with age. a** Immunoblot analysis of proteins from left ventricular lysate of WT mice on postnatal days (PD) 15, 60, 120. Quantification of immunoblots show that **b** CAP2 (***p = 0.0004, ****p < 0.0001, F = 45.87) and **c** α-smooth muscle actin (α-SMA) levels (*p = 0.0363, *p = 0.0337, F = 6.085) decrease with age, whereas **d** α-skeletal muscle actin (α-SKA) follows a biphasic expression pattern (***p = 0.0005, *p = 0.0250, F = 20.61) and **e** cActin is stably expressed (n.s.: p = 0.2429, n.s.: p = 0.5915, F = 1.529) compared to PD 15 (mean ± SD, n = 3–4 mice, one-way ANOVA, df = 2, n.s.: not significant).

inhibits actin polymerization and incorporation into thin filaments at the pointed ends and it depolymerizes actin filaments, as well as delays (inhibits) actin assembly.

Our FRAP experiments revealed that excess levels of CAP2 inhibited the rate of actin incorporation into thin filaments, whereas *Cap2*-null cardiomyocytes exhibited accelerated actin dynamics. Fitting the fluorescence recovery data to a two-phase exponential equation revealed a fast-phase and a slow-phase of actin recovery which are hypothesized to represent a highly dynamic and a more stable population of actin (potentially stabilized by Tpm and/or Tmod1), respectively[15,38,50]. Excess levels of CAP2 strongly decreased the rate of a highly dynamic population of actin only at the pointed ends. CAP2 affects the kinetics of actin assembly (rate and half-life) but not the mobile fraction of actin, especially in the slow phase, which is an indicator of the amount of actin assembled into the filaments. This result agrees with our finding that CAP2 has no direct effect on TFLs, since CAP2 does not completely cap and prevent actin from assembling into filaments but instead slows down its incorporation. Excess levels of Lmod2, another pointed end-binding protein in cardiomyocytes, increases the mobile fraction of actin in the slow (more stable population of filaments), but not in the fast phase[15]. Furthermore, Lmod2 does not affect the rate of actin assembly, but instead promotes incorporation of additional actin monomers onto thin filament pointed ends in cardiomyocytes. Lmod2 belongs to the Tmod protein family and possesses a Tpm-binding site[51,52], which increases its affinity for the pointed end[9,40]. Lmod2 competes with Tmod1 for binding at pointed ends to release Tmod1 from being an effective "cap" and allow filament elongation, which in turn leads to longer TFLs[10,52]. The differences observed in our FRAP results between CAP2 and Lmod2 indicate that they associate with different populations of actin filaments or alternatively, and that they influence actin polymerization at different stages of filament assembly. Together our

observations now add CAP2 to the small group of actin filament regulatory proteins, including Tmod1 and Lmod2, which influence actin polymerization and dynamics at thin filament pointed ends in the heart.

Actin incorporation into thin filaments is more dynamic in *Cap2*-KO cardiomyocytes, since they exhibited an increased rate constant and decreased half-life of actin assembly. Interestingly, although excess levels of CAP2 only appeared to alter actin assembly at the pointed ends, we found actin incorporation to be enhanced at both the barbed and the pointed ends in *Cap2*-KO cells. This result was surprising, since we did not detect the presence of endogenous CAP2 or GFP-CAP2 at thin filament barbed ends (near the Z-disc) in cardiac muscle by immunofluorescence staining. Therefore, we predict that CAP2 likely does not directly interact with thin filament barbed ends. An increase in actin dynamics at the barbed ends of *Cap2*-KO cardiomyocytes may be explained by interactions between CAP2 and G-actin. Biochemically, CAP2 is capable of sequestering G-actin and preventing it from polymerizing[16]. It is known that CAP1, the nonmuscle mammalian isoform, is highly abundant in the cytoplasm of nonmuscle cells, with a 1:4 molar ratio to actin[53]. Our immunofluorescence analysis indicates that CAP2 is present at the pointed ends and also diffusely in the cytosol of cardiomyocytes, consistent with its interaction with cytosolic G-actin. This suggests that the lack of monomer sequestering in *Cap2*-KO cardiomyocytes can lead to increased subunit addition to both ends of actin filaments. Therefore, we propose that in addition to its role at the pointed ends, CAP2's G-actin-sequestering ability is relevant to its physiological function. Together, these observations support the idea that CAP2 is able to associate with free actin molecules and inhibit monomer addition onto thin filaments. An alternative explanation for increased recovery of actin at the barbed ends of *Cap2*-KO cells could be due to CAP2's inhibitory role (that was discovered in vitro) on the actin polymerization

activity of inverted formin-2 (INF2), a barbed-end interacting protein[54]. However, this interaction has yet to be determined in muscle cells.

Although the ability of CAP2 to inhibit incorporation of actin into thin filaments is reminiscent of the capping function of Tmod1, surprisingly, we found that CAP2 is not effective in altering TFLs. One explanation for this difference is that CAP2 does not interact with Tpm. The affinity of Tmod1 for the pointed end dramatically increases by interacting with two Tpm molecules, and it sterically prevents the addition and loss of actin subunits[11,55,56]. The lack of a Tpm-binding site within CAP2 would be predicted to prevent it from interacting tightly with pointed ends of mature filaments and directly alter TFLs. Tpm's ability to instead antagonize the depolymerization activity of CAP2 further highlights the functional differences between these two proteins. These observations indicate that CAP2 and Tmod1 both interact with pointed ends, albeit differently, and potentially associate with thin filaments that possess distinct dynamic properties. Although varying the amount of CAP2 does not alter TFLs, we found it to affect the levels of G-actin and F-actin in cells. Based on this observation, we predict that unlike Tmod1 and Lmod2, CAP2 does not "fine-tune" the lengths of existing filaments but it plays a role in the disassembly of entire filaments.

The expression pattern of CAP2 indicates that it assembles early, while Tmod1 assembles into sarcomeres late in myofibrillogenesis, only after other thin filament proteins (α-actinin, actin, and Tpm) are assembled[7]. We found Cap2-null cardiomyocytes exhibit non-striated (i.e., immature) organization of thin filament-associated proteins, including actin and α-actinin, at earlier stages of myofibrillogenesis, indicating delayed cardiomyocyte differentiation and maturation. This phenotype is strikingly distinct from the absence of Tmod1, since Tmod1-KO cardiomyocytes contain fewer and narrower actin filaments[57]. Cap2-KO cardiomyocytes demonstrate an impaired α-actin isoform switch, since they exhibit high levels of α-SMA and α-SKA incorporated into thin filaments. Tmod1 has not been reported to be a factor involved in α-actin exchange and it appears that this function may be a unique property of CAP2 that is distinct from Tmod1's role in the heart. Therefore, we propose that CAP2 modulates α-actin exchange at the pointed ends and isoform distribution in thin filaments during cardiomyocyte differentiation.

How actin and Tpm can organize into mature myofibrils during myofibrillogenesis in the absence of stabilization by Tmod has remained an unsolved mystery [for review, see ref. [7]]. Since CAP2 is robustly expressed during embryogenesis[53] and possesses distinct functions (i.e., enhancing actin disassembly and α-actin exchange) from Tmod1, we predict that CAP2 is the long sought after "missing-link" in our understanding of thin filament assembly. Since non-striated myofibrils are reported to be precursors to striated myofibrils[58,59], it is likely that CAP2 plays an important role in the assembly of nascent thin filaments in the absence of Tmod1 by interacting with pointed ends and/or by limiting the levels of actin incorporation through its depolymerization activity, early in striated muscle development. This important CAP2 function in muscle development is evident in vivo; Cap2-KO mice demonstrate disease pathology as early as postnatal day 15 and succumb to sudden cardiac death[17,18,60]. In normal cardiac muscle, α-SMA is abundantly expressed in embryonic and neonatal hearts, and is replaced by cActin and α-SKA[46–48]. Cap2-KO hearts display impaired α-actin exchange, since cActin is down-regulated and substituted by incorporation of α-SMA and α-SKA into thin filaments. The α-actin switch in skeletal muscle of Cap2-KO mice is delayed during myofibril differentiation but takes place successfully in adult mice[19]. However, we found that this switch never occurs in Cap2-KO

hearts, even in adulthood. These findings are consistent with the observation that lethality of Cap2-KO mice is attributed to cardiac, but not skeletal deficits[17–19,60]. Therefore, we propose that CAP2 is more influential and indispensable for α-actin exchange and sarcomeric remodeling in cardiac muscle compared with skeletal muscle.

CAP associates with ADF/cofilin-decorated actin filaments and greatly accelerates their depolymerization rate[31,35]. We found that CAP2 increases G-actin and decreases F-actin levels, which is consistent with its proposed depolymerization activity. In addition, Cap2-KO cardiomyocytes contain higher levels of α-SMA and α-SKA, indicative of impaired actin clearance, potentially arising from perturbed depolymerization activity without CAP2. Cofilin-2 (CFL2), the muscle-specific isoform of ADF/cofilins, has also been proposed as a factor involved in disassembly of actin filaments near the M-line, in proximity to the pointed ends[61]. Intriguingly, skeletal muscle from Cfl2-KO mice also exhibit impaired α-actin switch[62]. Therefore, a synergy between CAP2 and CFL2 potentially exists in striated muscle with respect to regulating efficient depolymerization of actin filaments. In striated muscle, actin filaments are strongly protected against disassembly by regulatory proteins, including Tpm. Tpm1.1 inhibits the depolymerization activity of CFL2[63] and also CAP2 (shown above). Therefore, we hypothesize that CAP2 is responsible for accelerating cofilin-mediated depolymerization of a dynamic population of non-regulated actin filaments, potentially comprised of α-SMA and α-SKA, that are not stabilized by Tpm at their pointed ends. We predict that the absence of CAP2 in cardiomyocytes prevents the CAP2/cofilin-2 complex from forming and effectively "cleaning up" non-striated/regulated myofibrils. The high abundance of α-SMA and α-SKA-containing filaments in turn, compromises myocyte spreading and maturation.

Cap2-KO mice exhibit, DCM, ventricular arrhythmias and cardiac conduction problems[17,18]. A deleterious mutation in the CAP2 gene of human patients also leads to supraventricular tachycardia and severe DCM[23]. It is widely accepted that proper actin polymerization and expression levels of actin isoforms influence force development and contractility in muscle cells[64–67]. Furthermore, α-SMA is specifically involved in the beating frequency of embryonic cardiomyocytes[66] and increased expression of α-SKA in mice results in hyperdynamic hearts[67]. Similar to what is observed in Cap2-KO mice, the deficiency of CAP2 in human patients could result in compromised α-actin switch and isoform composition, leading to dysregulation of actin-thin filaments and impaired contractility of cardiac muscle. Inhibition of the serum-response factor, which regulates α-SMA expression, delays DCM onset in Cap2-KO mice[60]. In line with these observations, targeting α-actin expression is a potentially promising therapeutic approach for preventing DCM development and contractility deficits in the absence of CAP2.

## Methods

**Construct preparation.** Mouse CAP2 (mCAP2) and mouse Tmod1 (mTmod1) were cloned from cDNA generated from mouse hearts. Mouse cofilin-2 (mCFL2) cDNA clone (MG551672-G) was obtained from Sino Biological, Inc. (Wayne, PA). Rat striated muscle α-Tpm (rTpm1.1) in pET11d expression plasmid was a generous gift from Dr. Alla Kostyukova (Washington State University, Pullman, WA). mCAP2 was cloned into pEGFP-C2 vector (Clontech, Mountain View, CA) or into a modified pEGFP-C2 vector in which GFP was replaced with mCherry to create GFP-CAP2 or mCherry-CAP2 expression plasmids, respectively. GFP-cActin and mCherry-Tpm1.1 were cloned as described previously[15,68]. For recombinant protein expression, mCAP2, mTmod1, or mCFL2 with an N-terminal 6xHistidine-tag was cloned into pReceiver-B01 (GeneCopoeia, Rockville, MD).

**Protein expression and purification.** Globular actin (G-actin) was purified from actin acetone powder[69] obtained from rabbit skeletal muscle as described else

previously[70]. Recombinant Tpm1.1 with Ala-Ser extension on its N-terminus to mimic acetylation of Tpm was purified as described previously[71].

His-tagged mCAP2, mTmod1, or mCFL2 constructs were transformed into Rosetta™ 2 (DE3) pLysS competent cells (EMD Millipore) and grown to OD 0.6 in LB medium at 37 °C, 250 rpm. Protein expression was then induced with 0.1 mM IPTG at room temperature for 5 h. Cells were spun down at 5500 × g, 4 °C for 20 min. Cell pellets were resuspended in B-PER bacterial protein extraction reagent (Thermo Fisher Scientific, Waltham, MA) with DNase I, 1× Halt protease inhibitor cocktail (Thermo Fisher Scientific) and 0.1 mg/mL lysozyme (Sigma-Aldrich), and then incubated for 20 min at room temperature followed by 20 min on ice. The cells were sonicated on ice and the extract was clarified by centrifugation at 16,000 RPM (Beckman-Coulter, JA-17 rotor), 30 min, 4 °C. The supernatant of His-mTmod1 or His-mCFL2 was applied to Ni-NTA Superflow column (Qiagen, Valencia, CA) after equilibration with 50 mM sodium phosphate, (pH 8.0 for mTmod1 and pH 7.0 for mCFL2), 300 mM NaCl, 10 mM Imidazole (column buffer). Following washes with column buffer, His-mTmod1 or His-mCFL2 was eluted with column buffer containing 250 mM Imidazole. His-mTmod1 was dialyzed into 20 mM Tris–HCl, pH 8.0, 150 mM NaCl, 1 mM EDTA and His-mCFL2 was dialyzed into 10 mM MOPS [3-(N-morpholino)propanesulfonic acid], pH 7.0, 50 mM NaCl.

His-mCAP2 was observed in inclusion bodies after induction and the protein was recovered from inclusion bodies by resuspending the cell pellet in column buffer (pH 8.0) with 8 M urea. Solubilized protein was purified as described above, except under denaturing conditions in the presence of 8 M urea. Following elution, mCAP2 was dialyzed stepwise against 8 M urea in 20 mM Tris–HCl pH 8.5, 150 mM NaCl, 1 mM EDTA as in ref. [71] and concentrated using Amicon® Ultra-4 centrifugal devices (EMD Millipore).

All purified proteins were clarified by ultracentrifugation at 100,000 RPM (Beckman-Coulter TLA 120.1), 4 °C for 1 h and protein concentrations were measured by Pierce BCA protein assay kit (Thermo Fisher Scientific).

**Actin depolymerization assays**. 5 µM G-actin was polymerized into F-actin in F-buffer (25 mM Imidazole, pH 7.8, 100 mM KCl, 2 mM MgCl₂, 1 mM EGTA, 2 mM Tris–HCl, 0.2 mM CaCl₂, 0.2 mM ATP and 0.5 mM DTT) for 1 h, at room temperature. 5 µM Tpm1.1 was added to the filaments and incubated for 30 min, at room temperature. Depolymerization of F-actin in the absence or presence of Tpm1.1 was initiated by diluting actin filaments 5-fold with F-buffer in the presence of 2 µM CFL2 and differing concentrations of CAP2 (0, 0.03, 0.06, 0.125, 0.25, 0.5 µM). After incubation at room temperature for 30 min, the filaments were pelleted at 70,000 RPM (Beckman-Coulter TLA-100 Rotor), for 30 min, at room temperature. The supernatants and pellets were separated and solubilized in 1× Laemmli sample buffer for 30 min, at room temperature. The samples were boiled at 100 °C, 5 min and resolved on 11% SDS polyacrylamide gels. Gels were stained with Coomassie Brilliant Blue R-250 (Bio-Rad) and images were captured with an Epson Perfection 2450 Scanner. Quantitative analysis of the background-corrected band densities of the pellet and supernatant samples was done using ImageJ.

**Nondenaturing-PAGE**. 3 µM CAP2 or Tmod1 was allowed to interact with 6 µM Tpm1.1 or 3 µM actin in 20 mM Tris–HCl pH 8.5, 150 mM NaCl for 30 min, at room temperature. Individual proteins and their complexes were resolved using nondenaturing-PAGE as described previously[71]. The experiment was performed three times to ensure the reproducibility of results.

**Cap2-KO mice**. Embryos heterozygous for gene-targeted mice (Cap2 tm1a (EUCOMM)Wtsi) were obtained from the European Conditional Mouse Mutagenesis Program (EUCOMM) (EMMA ID: 05552). The embryos were re-derived in the Genetically Engineered Transgenic Core at the University of Arizona, and a colony was established by crossing to a C57BL/6J strain (Jackson Laboratory stock no. 000664). Genomic DNA was isolated by digesting tail tips in 100 µL digestion buffer (10 mM Tris–HCl, pH 8.3, 50 mM KCl, 1.5 mM MgCl₂, 0.45% NP-40, 0.45% Tween-20) plus ~6 U proteinase K (Thermo Fisher Scientific) at 60 °C, 20 min and then at 100 °C, 3 min. Genotyping primers adapted from ref. [17] were synthesized by Sigma-Aldrich. PCR conditions were as follows: 95 °C for 2 min, then 95 °C for 30 s/60 °C for 30 s/72 °C for 45 s repeated 39 times, then 72 °C for 5 min. Both male and female WT or Cap2-KO mice were investigated for body weight and stroke volume measurements. Male mice were used for thin filament length measurements, immunofluorescence, and immunoblot analysis, since the disease phenotype is more apparent in male mice as reported previously[17,18,60] and in this study. Work with animals was performed under the approval by The Institutional Animal Care and Use Committee at the University of Arizona; Protocol number 08-017, which confirmed to all applicable federal and institutional policies, procedures, and regulations, including the PHS Policy on Humane Care and Use of Laboratory Animals, USDA regulations (9 CFR Parts 1, 2, 3), the Federal Animal Welfare Act (7 USC 2131 et. Seq.), the Guide for the Care and Use of Laboratory Animals, and all relevant institutional regulations and policies regarding animal care and use at the University of Arizona.

**Cell isolation, culture, and transduction**. Murine cardiomyocytes were isolated from postnatal day 3 (PD 3) or younger mixed gender Sprague-Dawley rats or

C57BL/6J mice as described[15]. Chick cardiomyocytes were isolated from day 6 embryonic chick hearts and plate as described[7]. Isolated cells were plated in 35 mm dishes on 12 mm-diameter Matrigel-coated (1:1000) number 1.5 coverslips for staining or gelatin-coated (2%) dishes for collecting lysate. For super-resolution structured illumination microscopy (SR-SIM) experiments, chick cardiomyocytes were plated on Matrigel-coated (1:1000) high precision number 1.5 coverslips (0.170 ± 0.005 mm, Bioscience Tools).

Adenovirus expressing GFP, GFP-mCAP2, GFP-cardiac actin, GFP-mTmod1, GFP-mLmod2, mCherry, mCherry-CAP2 or mCherry-Tpm1.1 was constructed as described[15]. Cardiomyocytes were transduced with adenovirus expressing GFP-tagged or mCherry-tagged proteins. Multiplicity of infection (MOI) used for each viral construct was determined empirically, where >95% of cardiomyocytes were transfected, as determined by GFP-positive or mCherry-positive cells. Cells were transduced 1–2 days after plating with GFP-cardiac actin, GFP-alone, GFP-mCAP2, GFP-mLmod2 or mCherry-Tpm1.1 (5 MOI) and GFP-mTmod1, mCherry-alone or mCherry-CAP2 (20 MOI) adenovirus.

Cells were not specifically tested against mycoplasma contamination. However, for each culture cells were monitored by light microscopy daily and if contamination was suspected (based on observation of small, circular cells, and subsequent disruption of cardiomyocyte beating), the cultures were discarded.

**Lat A treatment of cells**. 2 days after plating, cells were incubated with 0.01% dimethyl sulfoxide (DMSO, control) or 2 µM Lat A (Invitrogen) at 37 °C, 30 min. Cells were washed with 1×PBS twice and fixed with 2% (wt/vol) paraformaldehyde in 1×PBS for 20 min. Cells were considered to have thin filament pointed-end assembly of CAP2/Tmod1, if a clear striation was visible in the majority of sarcomeres within the cell.

**Immunostaining**. A section of LV free wall was fixed overnight in 4% (wt/vol) paraformaldehyde/PBS, washed extensively in PBS, embedded in Tissue-Tek O.C.T. compound (Sakura Finetek), and frozen immediately in 2-methylbutane cooled in liquid N₂. 5-µm cryosections were cut and mounted onto number 1.5 coverslips.

2–3 days after transduction, cells were washed with 1×PBS twice and fixed with 2% (wt/vol) paraformaldehyde in 1×PBS for 20 min, at room temperature. Cells or cryosections were permeabilized in 0.2% Triton X-100 in 1×PBS for 20 min at room temperature, blocked with 2% BSA, 1% normal donkey serum in 1×PBS for 1 h at room temperature, and incubated overnight at 4 °C with primary antibodies diluted in 2% BSA, 1% normal donkey serum in 1×PBS, which included: rabbit polyclonal anti-CAP2 (1:100) (15865-1-AP, Proteintech, Rosemont, IL), mouse monoclonal anti-α-actinin (1:200) (EA-53, Sigma-Aldrich, St. Louis, MO), rabbit polyclonal anti-Tmod1 (2 µg/mL)[72], mouse monoclonal anti-myomesin (1:100) (B4, a kind gift from Dr. Elizabeth Ehler, King's College, London, UK), mouse monoclonal anti-α-smooth-muscle-actin-FITC (1:300) (clone 1A4, Sigma-Aldrich), mouse monoclonal anti-α-skeletal-muscle-actin (1:100) (MUB0108P, Exalpha Biologicals Inc.), rabbit polyclonal anti-myosin binding protein-C (1:200) (Myomedix, Germany), rabbit polyclonal anti-desmin (1:20) (Biomeda, Foster City, CA) and mouse monoclonal nonmuscle myosin IIB (CMII 25, Developmental Studies Hybridoma Bank).

Coverslips were then washed with 1×TBST for 5 × 5 min, and incubated for 2 h at room temperature with secondary antibodies (all from Thermo Fisher Scientific) diluted in 1×TBST, which included Alexa Fluor 405-conjugated goat anti-mouse IgG (1:200), Alexa Fluor 488-conjugated goat anti-rabbit IgG (1:500), Alexa Fluor 488-conjugated donkey anti-mouse IgG (1:500), Alexa Fluor 594-conjugated goat anti-rabbit IgG (1:300), Alexa Fluor 594-conjugated goat anti-mouse IgG (1:300), and Texas Red™-X Phalloidin (1:50). Coverslips were then washed with 1×TBST for 5 × 5 min and mounted onto slides with Aqua Poly/Mount (Polysciences Inc., Warrington, PA) for deconvolution microscopy or ProLong Diamond Antifade Mountant (Thermo Fisher Scientific) for super-resolution microscopy.

Images of cultured cardiomyocytes were captured using a Nikon Eclipse Ti microscope with a ×10 NA 0.3 or ×100 NA 1.5 objective, and a digital CMOS camera (ORCA-flash4.0, Hamamatsu Photonics, Shizuoka Prefecture, Japan). 3D deconvolution was performed using NIS offline deconvolution software (Nikon Corporation, Tokyo, Japan) and images were processed with ImageJ. Super resolution microscopy was performed using a Zeiss ELYRA S1 (SR-SIM) microscope equipped with an AXIO Observer Z1 inverted microscope stand with transmitted (HAL), UV (HBO) and solid-state (405/488/561 nm) laser illumination sources, a ×60 objective (NA 1.45), and EM-CCD camera (Andor iXon). ZEN software (Zeiss) was used to acquire the images, to perform structured illumination, and to analyze the peak profiles of line scans. Images of LV tissue sections were captured using a Deltavision RT system (Applied Precision) with a ×100 NA 1.3 objective and a CCD camera (CoolSNAP HQ; Photometrics). Images were deconvolved using SoftWoRx software and processed using ImageJ.

Fluorescence imaging was conducted with cells from at least three independent cultures or animals with 15–30 cells investigated per culture or animal. Representative images were selected based on the staining pattern observed in the majority of cells.

**Thin filament length measurements**. GFP or GFP-CAP2-expressing rat cardiomyocytes and WT or Cap2-KO mouse cardiomyocytes were washed with 1×PBS

twice and incubated in relaxing buffer (10 mM MOPS, pH 7.4, 150 mM KCl, 5 mM MgCl$_2$, 1 mM EGTA, and 4 mM ATP) for 15 min and fixed with 2% paraformaldehyde in relaxing buffer for 15 min, at room temperature. TFLs and sarcomere lengths were measured using the DDecon plugin for Image J (Littlefield and Fowler 2002; Gokhin and Fowler 2017). Line scan measurements of cActin staining from the LV tissue of PD 120 WT or Cap2-KO mice was performed using ImageJ.

**Fractionation of cellular G-actin and F-actin**. Monomeric and filamentous actin populations in murine cardiomyocytes were separated by using G-Actin/F-actin In Vivo Assay Biochem Kit (Cytoskeleton, Inc.) according to manufacturer's recommendations. Briefly, 5 days after plating cells were washed twice with 1×PBS and scraped off from gelatin-coated (2%) 35 mm plates in Lysis and F-actin Stabilization Buffer prewarmed to 37 °C. The solubilized lysates were incubated at 37 °C, 10 min and cell debris was clarified by centrifugation at $350 \times g$, 5 min. G-actin in the supernatants and F-actin in the pellets were fractionated by ultracentrifugation at $100,000 \times g$, 37 °C, 1 h. The pellets were solubilized in F-actin Depolymerizing Buffer on ice, 1 h by triturating with pipetting every 15 min. Supernatant and pellet fractions were resolved on SDS–PAGE, and actin in each fraction was detected by immunoblotting.

The lysis conditions were confirmed by treating the cells 4 days after plating with 0.01% DMSO (control), 2 μM Lat A (Invitrogen), or 0.15 μM jasplakinolide (Santa Cruz Biotechnology) overnight, at 37 °C and harvesting the lysates next day as described above.

**Pull-down assays**. NRCMs were plated onto gelatin-coated (2%) 60 mm dishes and transduced with adenovirus expressing GFP, GFP-Lmod2, GFP-Tmod1, and GFP-CAP2 2 days after plating. Five days after plating, cells were collected in ice cold IP buffer (20 mM Tris–HCl, pH 8.0, 150 mM NaCl, 1% NP-40, 2 mM EDTA, 10% glycerol, and 1× Halt protease inhibitor cocktail [Thermo Fisher Scientific]), sonicated on ice and spun down at $16,000 \times g$ for 15 min at 4 °C. The supernatant was collected and protein content was determined by the Pierce BCA protein assay kit (Thermo Fisher Scientific). 0.35 mg/mL of lysate was incubated overnight at 4 °C on GFP-affinity beads (a generous gift from Dr. Gregory Rogers, University of Arizona). The next day, GFP-beads were spun at $2500 \times g$ for 3 min at 4 °C and the supernatant was subsequently removed. Three washes were performed with ice cold IP buffer, with samples spun at $2500 \times g$ for 3 min, in between each wash. Samples were then boiled in 1× Laemmli sample buffer at 100 °C for 10 min, resolved on a 10% SDS polyacrylamide gel, and transferred to a PVDF membrane (0.45 μm; Fisher Scientific) for immunoblotting.

**Immunoblotting**. Rat or mouse cardiomyocytes were washed twice with 1×PBS and scraped off from gelatin-coated (2%) 35 mm plates in ice cold lysis buffer (25 mM HEPES, pH 7.4, 150 mM NaCl, 1.5 mM MgCl$_2$, 1 mM EGTA, 10 mM sodium pyrophosphate, 10 mM sodium fluoride, 0.1 mM sodium deoxycholate, 1% Triton X-100, 1% SDS, 10% (vol/vol) glycerol, 1× Halt protease inhibitor). Cell suspensions were sonicated on ice and spun down for 15 min at $16,000 \times g$ at 4 °C. A section of LV free wall (10–30 mg) was placed in a microcentrifuge tube containing 300 μL of ice-cold lysis and 200 mg of stainless steel beads (0.9–2.0 mm diameter blend; Next Advance Inc.) and was homogenized in a Bullet Blender (BBX24; Next Advance Inc.) at speed 10 for 4 min at 4 °C. Samples were spun down for clarification for 15 min at $16,000 \times g$ at 4 °C. Protein concentration was normalized by Pierce BCA protein assay kit (Thermo Fisher Scientific). Samples were boiled in 1× Laemmli sample buffer at 100 °C for 5 min, resolved on 8%, 10%, or 12% SDS polyacrylamide gels, and transferred to a nitrocellulose membrane (0.2 μm; Amersham Protran GE Healthcare). Total lane density of transferred proteins stained with Ponceau S was used to control for loading/transfer differences. The membranes were blocked in 5% (wt/vol) nonfat dried milk in 1×TBS for 1 h at room temperature and incubated with primary antibodies diluted in 2% BSA in 1×TBST overnight at 4 °C. Primary antibodies included rabbit polyclonal anti-CAP2 (1:2000) (15865-1-AP, Proteintech, Rosemont, IL), rabbit polyclonal anti-Tmod1 (0.2 μg/mL)[72], and rabbit polyclonal anti-Lmod2 (1:1000) (E13, Santa Cruz Biotechnology, Dallas, TX), mouse monoclonal anti-cardiac actin (1:1000) (03-61075, American Research Products, Inc., Waltham, MA), goat polyclonal anti-α-actinin (1:400) (AF8279, R&D Systems, Minneapolis, MN), mouse monoclonal anti-GFP (1:5000) (B-2, Santa Cruz Biotechnology), mouse monoclonal anti-α-smooth-muscle-actin (1:2000) (A-2547, Sigma-Aldrich), mouse monoclonal anti-α-skeletal-muscle-actin (1:2000) (MUB0108P, Exalpha Biologicals Inc.), mouse monoclonal anti-α-muscle-actin (1:2000) (MUB0107P, Exalpha Biologicals Inc.), mouse monoclonal anti-Tpm-1 (1:1000) (TM311, Novus Biologicals), rabbit polyclonal anti-pan actin (1:1000) (AAN01-A, Cytoskeleton, Inc.) and mouse monoclonal anti-GAPDH (0.9 μg/mL) (clone 6C5; Life Technologies). Following 5×10 min 1×TBST washes, the membranes were incubated with fluorescently labeled secondary antibodies including Alexa Fluor 680 or Alexa Fluor 790 AffiniPure anti-rabbit, anti-mouse or anti-goat IgG (1:40,000; Jackson ImmunoResearch) diluted in 5% (wt/vol) nonfat dried milk in 1 × TBST at room temperature for 2 h. Following 5 × 10 min 1 × TBST washes and 1 × 10 min 1 × TBS wash, the blots were imaged and analyzed using a LI-COR Odyssey CLx imaging system (LI-COR Biosciences, Lincoln, NE).

For validation, several antibody products were tested against each protein. Antibodies were first picked based on the manufacturers' recommendations on cross-reactivity and specificity. We further validated the antibodies by testing them by both immunostaining or immunoblotting. We observed certain antibodies to produce signal in one application but not in the other, therefore we limited their use to certain applications. Known molecular weights of detected proteins were used as an indicator of specificity in immunoblots. For immunostaining of proteins that have known cellular localizations, we compared our findings to published data to validate our results. Secondary antibodies alone were always used to determine the background signal. Antibodies against CAP2, Lmod2, and Tmod1 did not produce any signal in the tissues of the respective KO mouse models for these proteins by both immunofluorescence and immunoblotting.

**Fluorescence recovery after photobleaching**. A Leica SP5-II confocal microscope with a ×63 NA 1.4 objective and a 488-nm argon laser was used for FRAP experiments. Cardiomyocytes were plated on Matrigel-coated (1:1000), glass-bottomed dishes (MatTek), and were maintained at 37 °C and ∼5% CO$_2$ for the duration of the experiment. Three prebleach images were recorded, followed by photobleaching for ∼2 s at 80% total laser power. To monitor recovery after photobleaching, images were captured in successive intervals of 1 s (for a duration of 30 s), 5 s (for a duration of 150 s), and 10 s (for a duration of 600 s). Images were imported into Leica LASX software and analyzed as described previously[15]. Mobile fractions and halftimes were determined from nonlinear regression curves best fit using a two-exponential association equation $R = M_{fast} \times [1 - \exp(-k_{fast} \times t)] + M_{slow} \times [1 - \exp(-k_{slow} \times t)]$ with Prism 8.4.1 software (GraphPad Software, Inc., San Diego, CA). $R$ is the relative recovery at time $t$, $M$ is the mobile fraction and k is the rate constant. Half times ($t_{fast}$ or $t_{slow}$) are 0.69/$k$. Recovery was determined independently at three barbed and pointed thin filament ends per cell. If fitting the two-phase exponential equation to the fluorescence recovery data did not provide an $R^2$ value > 0.70 or did not converge, the calculated parameters were deemed unreliable and not included in analysis.

**Echocardiography**. PD 15, 60 or 120 mice were placed in dorsal recumbence on a heated (37 °C) platform for echocardiography. Transthoracic echocardiographic images were obtained with Vevo 2100 High-resolution Imaging System (VisualSonics) using model 707B or MS-550D transducer arrays. Images were collected and stored as a digital cine loop for off-line calculations. Standard imaging planes and functional calculations were obtained according to American Society of Echocardiography guidelines. M-mode images at the level of the papillary muscles were used to determine LV stroke volume.

**Statistics and reproducibility**. All statistical analyses were performed in Prism 8.4.1 (GraphPad Software, Inc., San Diego, CA). Normal distribution of the data was confirmed by Q–Q plots. Two-tailed Student's $t$-test, one-way ANOVA, two-way ANOVA with Tukey's post-hoc test or Log-rank (Mantel–Cox) test was used depending on the number of groups or variables. Confidence interval level was 95% for multiple comparisons. Differences with $p < 0.05$ were considered statistically significant. 3–4 independent experiments or cultures were performed for recombinant protein work or cell culture work, respectively. Samples or organisms were allocated into groups based on treatment, genotype, gender, and age. For cell culture experiments, control treated (buffer, DMSO, GFP or mCherry) cells were grouped and compared to the treatment of interest. For animal experiments, wild-type or KO mice were categorized separately and comparisons were performed only within the same gender and age. Data from an individual culture/experiment was defined as a replicate. For FRAP and TFL experiments, each measurement from a single cell was defined as a replicate, since the cells are derived from 10 to 15 neonatal rat pups per culture. For animal work, 6–8 mice per genotype per gender were used. Data with a sample size >8 mice resulted from taking measurements at different ages from the mice grown to PD 120. We chose not to discard any collected data, therefore the sample size varied in experiments involving mice. The only data exclusion was done in FRAP and thin filament length measurements. If fitting the two-phase exponential equation to the fluorescence recovery data did not provide an $R^2$ value > 0.70 or did not converge, the calculated parameters were deemed unreliable and not included in analysis. Similarly, for thin filament length measurements, if the fit equation did not converge that data set was discarded. All attempts of replication were successful as demonstrated by dot-plot format in the figures.

If the treatment type of a subject could be distinguished (i.e. genotyping in mice) after data collection and analysis, the groups were blinded. When applicable, folders containing data sets were blinded and revealed after analysis (i.e. FRAP and thin filament length measurements). For certain applications (immunoblots and immunofluorescence) blinding was not possible since the treatment (i.e. presence of GFP fluorescence or the CAP2 gene) had to be determined prior to collecting data.

**Data collection**. ImageJ Version 1.52 (NIH) was used for collecting data from SDS–PAGE or nondenaturing-PAGE gels and images of cardiac actin immunostain. ZEN software 2.3 Blue Edition (Zeiss) was used for acquiring super-resolution microscopy data. Image J Version 1.52 DDecon plug-in was used for thin filament

length measurements. Data from immunoblot membranes were collected using Image Studio Lite Version 5.2.5 (LI-COR Biosciences). FRAP data were acquired with Leica LAS X 3.4.2.18368. Echocardiography data were collected with Vevo Lab 3.2.6 (VisualSonics).

**Data analysis**. Prism 8.4.1 (GraphPad Software, Inc., San Diego, CA) was used for compiling the data, creating the figures and statistical analysis. Fluorescence recovery data from FRAP experiments were analyzed with Leica LAS X 3.4.2.18368. Echocardiography data were analyzed with Vevo Lab 3.2.6 (VisualSonics).

**Reporting summary**. Further information on research design is available in the Nature Research Reporting Summary linked to this article.

## Data availability
The datasets generated during and/or analyzed during the current study are available from the corresponding author on reasonable request. Source data for main figures can be found in Supplementary Data 1.

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

## Acknowledgements

We thank Rachel Mayfield for technical assistance with maintaining the rat and mice colonies, culturing of neonatal rat and mouse cardiomyocytes and echocardiography measurements, Christine Henderson for cloning mCAP2 into pEGFP-C2, Stefanie Novak for help with chick cardiomyocyte cultures, Miensheng Chu for help with adenovirus purification and Christopher Pappas for his help with FRAP experiments and careful reading of the manuscript. This work was supported by the University of Arizona Sarver Heart Center Marjorie Hornbeck Memorial Research Grant and American Heart Association Postdoctoral Fellowship 19POST34450023 (to M.C.); the University of Arizona Sarver Heart Center Finley and Florence Brown Endowed Research Award, NIH F30HL151139 and the Interdisciplinary Training in Cardiovascular Research (NHLBI) training grant T32HL007249 (to J.I.); and NIH Grants R01HL123078 and R01GM120137 and Czarina M. & Humberto S. Lopez Endowed Chair (to C.C.G.).

## Author contributions

C.C.G. and M.C. conceived the project. M.C., J.I. and C.C.G. designed the experiments. M.C. and J.I. performed the experiments, analyzed the data, and assembled the figures. M.C., J.I. and C.C.G. wrote the manuscript. All authors reviewed and approved the final version of the manuscript.

## Competing interests

The authors declare no competing interests.
