## [Peer Review File · Communications Biology]

Reviewers' comments:

Reviewer #1 (Remarks to the Author):

Summary:

In this work the authors characterize the role of cyclase-associated protein 2 (CAP2) in regulating the organization and dynamics of actin filaments that comprise sarcomeres within cardiomyocytes. This work is motivated by the previous findings that CAP2 localizes to M-lines within sarcomeres and is required for normal cardiac function in both mouse models and humans. Here, the authors provide mechanistic insight into the function of CAP2 within cardiomyocytes by showing that CAP2 localizes to filament pointed ends within sarcomeres, slows monomer incorporation into pointed ends, and antagonizes tropomyosin stabilization of F-actin by enhancing cofilin-mediated actin disassembly. Furthermore, the authors provide evidence that these CAP2 functions are required for cardiomyocyte maturation and the correct composition of alpha-actin in mature thin filaments.

Overall impression:

While the roles of CAP as an actin regulatory protein have been defined previously through in vitro experiments and work in model organisms (as the authors indicate in their introduction and discussion sections), this study adds to our understanding of how CAP contributes to sarcomeric actin organization in cardiomyocytes. Specifically, here it is shown for the first time that CAP2 localizes to the pointed ends of actin filaments in sarcomeres, regulates actin dynamics in cardiomyocytes, collaborates with cofilin to overcome tropomyosin stabilization of F-actin, and regulates actin isoform composition in cardiomyocytes. These novel findings pave the way for additional research into how actin disassembly factors in cardiomyocytes contribute to cardiomyocyte maturation. These findings are supported by their data, which is clearly presented and well explained. I have recommended minor revisions that largely pertain to the interpretation and presentation of the data.

Comments:

1) Comment: In the section titled: CAP2 associates with a dynamic population of actin filaments the authors treat cells with a low concentration of actin depolymerizing drug Lat A and observe a loss of CAP2 at filament pointed ends. They conclude, "This result suggests that assembly of CAP2 at the pointed end is dependent on active actin polymerization in cardiomyocytes." However, CAP2 also binds to G-actin making it seem possible that the increased free G-actin in the cytosol could compete CAP2 away from pointed ends. Suggested changes shown below:

- a. Is there evidence that Lat A disrupts the interaction between actin monomers and CAP2? If so, the authors should indicate this and provide the citation.
- b. If it is possible that increased monomeric actin is competing CAP2 away from actin pointed ends, this alternative interpretation should be added to their interpretation of the results in this section.

2) Comment: Figure S1 is titled: Excess levels of CAP2 has an insignificant effect on thin filament lengths. This is unclear because the differences appear statistically significant.

- a. Suggested change: Change the language to "minor effect" or another alternative phrase.

3) Comment: In Figure S1, filament lengths are shown in a bar graph that does not show individual data points. It would also make the data easier to interpret if the data were represented as a dot plot so that each datapoint is visible. For example, is there a population of filaments that have shorter lengths?

- a. Suggested change: Replace the bar graph with a graph that shows individual measurements.

4) Comment: In the sections CAP2 depolymerizes actin filaments and Tropomyosin inhibits the depolymerization activity of CAP2 the authors describe changes in the percentage of F-actin and G-

actin without providing the actual percentage measured or the fold change or the under the conditions investigated. This makes it difficult to follow the magnitude of the change in different experiments.

a. Suggested change: Instead of only saying "higher" or "lower" the authors should include the initial percent of G-actin measured and the fold change under different conditions.

5) Comment: In Figure 4E there is a 15 kDa size marker that needs to be moved to the left.

6) Comment: In Figure 4F, the label on the vertical axis should be changed to "Band density of F-actin in the pellet" to clarify what is being quantified on the gel.

7) Comment: In Figure 4 the authors conclude that "CAP2 depolymerizes actin filaments but stabilization by tropomyosin prevents CAP2-mediated filament disassembly." This is supported by the blot and accompanying graph in panels 4E and 4F. However, given that tropomyosin has been previously shown to stabilize actin filaments and protect them from disassembly by cofilin, this finding is not unexpected. The more interesting, and potentially relevant, finding is that CAP2 appears to help cofilin overcome the stabilization of F-actin by tropomyosin. This may be particularly relevant given that thin filaments in sarcomeres are decorated with tropomyosin. Suggested changes below:

a. In their assay, is there a concentration of CAP2 that when added to cofilin allows cofilin to disassemble tropomyosin decorated filaments to the same degree seen with undecorated filaments? This can be addressed in the text associated with figure 4 in section Tropomyosin inhibits the depolymerization activity of CAP2.

b. In the graph in figure 4F it appears that datapoints are normalized to the amount of actin in the pellet observed when cofilin is added. This needs to be clarified in the figure caption. Are the trends the same if all data points are normalized to the actin only condition?

8) Comment: In the section titled: Tpm1.1 recruits Tmod1, not CAP2 to thin filaments, the authors conclude "Therefore, Tpm1.1 favors the recruitment of Tmod1 (due to its known tropomyosin-binding sites) but not CAP2, to thin filaments, suggesting that CAP2 cannot directly bind to filaments stabilized by Tpm1.1." However, it is not clear why they come to this conclusion given that the fraction of CAP2 that pellets with F-actin is unchanged with increased Tpm1.1 expression (see Fig. 4J).

a. Suggested change: The authors should clarify how their data indicates CAP2 is incapable of interacting with tropomyosin decorated filaments or else change the wording in this section.

9) Comment: In figure 4B, the authors show that increased CAP2 expression shifts more actin into the monomer fraction and decreases F-actin. This seems inconsistent with the finding the TFL does not change.

a. Suggested change: The authors should add additional text to the discussion to address how it is that there is less actin in polymer form in CAP2 overexpressing cells but the overall TFL does not change (Fig. S1).

10) Comment: In the section titled: CAP2 inhibits incorporation of actin onto thin filament pointed ends. The authors show via FRAP that actin recovery at the pointed ends is diminished with CAP2 overexpression but the mobile fraction is unchanged. They state, "Although, the rate of actin incorporation at the pointed ends was diminished by CAP2, the mobile fraction at the slow phase of recovery was unaffected, indicating that the amount of actin assembled at the end of the experiment was unchanged between cells transduced with mCherry-CAP2 and mCherry." This is in agreement with their finding that TFL does not change with increased expression of CAP2 (Fig. S1). However it seems to disagree with their finding in figure 4B where they show that CAP2 overexpression results in a 10% increase in G-actin. The authors should make changes to the text to answer the questions below:

a. What is the difference between what is being measured in their pelleting assay and FRAP experiments such that the pelleting assay indicates CAP2 reduces monomer incorporation into polymer

while FRAP does not?

b. How are the authors accounting for recovery due to GFP labeled G-actin diffusing into the bleached area from the cytosol without being incorporated into F-actin?

c. If the concentration of free G-actin in the cytosol is different between their two different conditions (as they indicate in figure 4B) could this influence their measurements?

11) Comment: In figure 6 discussed in the section titled, "Cap2-null cardiomyocytes display altered α -actin assembly and expression", the authors show single representative images of different actin isoform staining for the WT and KO cells.

a. Suggested change: The authors should indicate in the text or the figure how repeatable these results are. How many cells were looked at? Did all cells display the same phenotype (disorganized sarcomeres, reduced staining, etc.)?

12) Comment: In the section titled: Cap2-null cardiomyocytes display altered α -actin assembly and expression, the authors state "Although cActin assembled into thin filaments in WT and Cap2-KO cells, its pattern was more irregular in Cap2-KO cardiomyocytes (Fig. 6D)." It is unclear what irregular means. Do the authors mean that there is higher cell-to-cell variability or variability within a single cell?

a. Suggested change: Clarify the description of the phenotype in the text.

13) Comment: In the example image shown in figure 6D, it appears that there is less cActin in the KO cells. Additionally, in the quantification of cActin by immunoblot in figure 6F, in two out of three cultures it appears there is less cActin in KO cells than in WT cells.

a. Suggested change: If the example image gives a false impression of the levels of cActin in KO cells, a different image should be selected.

b. Suggested change: If in some cultures of the KO line there is less cActin, this should be discussed in the text to indicate why there is culture-to-culture variability in the amount of cActin measured in KO cells. This may also be clarified by repeating the experiment to determine if one of the measurements is an outlier and that there is actually a trend to have less cActin in KO cells.

14) Comment: In the section titled: Cap2-null cardiomyocytes display altered α -actin assembly and expression, the authors state "Collectively, our results indicate that expression levels and assembly of α -SMA and α -SKA are significantly altered in Cap2- KO cardiomyocytes, indicating CAP2 plays an important role in the distribution of α -actin isoforms during cardiac muscle development." It is unclear what "distribution" means in this sentence.

a. Suggested change: The authors should change the text to clarify how CAP2 regulates the distribution of alpha-actin isoforms. Distribution between F-actin and G-actin? Distribution within a single filament?

15) Comment: "In the section α -SMA and α -SKA replace cActin in thin filaments from Cap2-KO mice", the authors state "Third, immunostaining of Cap2-KO hearts revealed that striations of cActin were narrower compared to WT hearts (Fig. 7D)." This is an interesting observation given that TFL does not change in Cap2-KO mice. The authors should expand on what this observation means by addressing the questions below:

a. Does this observation indicate that loss of CAP2 restricts cActin to a specific location within the thin filament? For example, is cActin enriched at the barbed or pointed ends of thin filaments in KO cells?

b. Are the other two actin isoforms uniformly distributed within thin filaments?

16) Comment: In the discussion section the authors state "An increase in actin dynamics at the barbed ends of Cap2-KO cardiomyocytes may be explained by interactions between CAP2 and G-actin." However, CAP has also been shown to inhibit the activity of the formin INF2 (DOI:

10.1038/s41556-019-0307-4 and DOI: 10.1073/pnas.1914072117). Additionally, INF2 is expressed in cardiomyocytes and localizes to Z-bands (DOI: 10.1091/mbc.E13-08-0443).

a. Suggested change: To address this the authors should indicate that increased barbed end assembly with loss of CAP2 may be due to increased INF2 actin-assembly activity and include the relevant citations indicated above.

17) Comment: The authors discuss the role of CAP2 in stabilizing actin giving contradictory conclusions in different sections. In the discussion section the authors describe their FRAP results stating, "Excess levels of CAP2 strongly stabilized the highly dynamic population of actin only at the pointed ends by decreasing the rate constant of recovery." However, in the section titled CAP2 depolymerizes actin filaments the authors state, "To decipher CAP2's specific function at thin filament pointed ends, we tested the effect of CAP2 on the stabilization of actin filaments... Collectively, this result is the first to hint at CAP2's distinct role in cardiac muscle: it promotes depolymerization of F-actin into G-actin and it is a new regulator of actin filament disassembly in cardiomyocytes."

a. Suggested change: The language in the manuscript, and specifically use of the word 'stabilize' should be made consistent. When the authors refer to actin stabilization do they mean decreasing loss of actin monomer from F-actin or decreasing the addition of new monomers or both?

Reviewer #2 (Remarks to the Author):

The uniform length of actin thin filaments is a remarkable feature of striated muscle. The mechanisms of precise assembly and length maintenance of actin filaments are still not completely understood. In particular, the molecular machinery involved in actin dynamics is not fully defined. In the submitted manuscript the Authors addressed the unresolved problem of the role of cyclase-associated protein 2 (CAP2) in maturation of the thin filaments in cardiomyocytes. A wide range of analyzes were carried out on rat neonatal cardiomyocytes, chick embryonic cardiomyocytes, and cardiomyocytes isolated from Cap2-KO mice. Using SR-SIM the Authors confirmed unequivocally that in cardiac muscle CAP2 binds to the pointed ends of actin filaments which undergo dynamic disassembly. CAP2 function was found to be different from the functions of two other pointed end-binding proteins – tropomodulin and leiomodin. Overexpression of CAP2 had little effects on the length of the filaments, but strongly affected the switch of actin isoforms, a process crucial for cardiomyocytes differentiation. The results of the CAP2 overexpression and knock out experiments were consistent. Each experiment was strictly controlled and well documented, an appropriate statistical analysis was applied. I am impressed with this thorough work, which unveils detailed molecular mechanisms of the thin filaments maturation in cardiac muscle.

There are few issues, which are not clear and I would like the Authors to comment on them.

1. In the experiment illustrated in Fig. 4 E-F the Authors showed that Tpm1.1 inhibited the acceleration of CFL2-induced actin depolymerization by CAP2. However, at the concentrations of cofilin used in this experiment there was practically no tropomyosin bound. How could tropomyosin protect actin from depolymerization if it was removed by cofilin? Also, in Fig. 4E the CFL2 -/+ row should be aligned with the appropriate gel bands.

2. I cannot agree with the Authors that tropomyosin reduces the binding of cofilin. In the Discussion section (page 10, last paragraph) they refer to some older work, which suggested such effects, but the recent studies on muscle tropomyosin isoforms did not confirm such mode of Cof-2 action (e.g. doi:10.3390/ijms21124285). Therefore, the hypothesis that CAP2 together with CFL2 is responsible for depolymerization of the population of actin filaments that are not stabilized by tropomyosin should

be revised.

3. Using various methods the Authors showed that CAP2 tends to associate with a more dynamic population of thin filaments. I must admit that it is not entirely clear to me what the population of dynamic filaments in cardiomyocytes is. Fig. 1C shows a perfect alignment of CAP2 and Tmod1. Does it mean that in sarcomere the I-bands contain a mixture of parallel thin filaments that are either stabilized by Tpm-Tmod or are Tpm-free, but capped by CAP2?

4. Although CAP2 is the main focus of these studies, in the Introduction the Authors have not included any information on CAP2 structure and modes of actin binding. A short description of the multidomain structure and oligomerization ability would give readers a better background and would help to understand the complex functions of CAP2 in cardiomyocytes.

5. Two recent papers have proposed models of CAP-driven dynamics of actin filaments (doi: 10.1038/s41467-019-13268-1 and doi: 10.1038/s41467-019-13213-2). In these papers yeast isoform Svr2/CAP and CAP1 were used. In the light of the data collected in the present study, can these models be applied to CAP2 function in cardiomyocytes?

Reviewer #3 (Remarks to the Author):

Summary

CAP2 is an isoform of cyclase-associated protein (CAP) that is expressed in muscle. Mutation of CAP2 causes defects in contractile apparatuses in cardiac and skeletal muscles in mice and cardiac disease in humans. However, cell biological function of CAP2 was largely unknown. This work presents excellent functional characterization of CAP2 in mouse cardiomyocytes and reports that CAP2 plays important roles in actin dynamics at the pointed ends of actin filaments in sarcomeres. They also report that CAP2 knockout causes aberrant up-regulation and sarcomere incorporation of smooth- and skeletal-muscle actin isoforms and inhibits maturation of cardiac myofibrils. CAP2 does not appear to affect the thin filament lengths and is highly expressed in developing hearts. Therefore, CAP has a distinct function from leiomodin and tropomodulin, which also regulates pointed-end actin dynamics in sarcomeres. The provided information significantly advances our understanding of the cell biological mechanism of sarcomere assembly. There are several relatively minor concerns that should be considered by the authors.

1. In Fig. 4E, the protective role of tropomyosin for CAP2 and CFL2 is shown by pelleting assays. The gels are only shown for the pellet fractions. However, gel images for the supernatant fractions should also be included. This is important to show the equal amounts of total actin in all the reactions and no protein degradation occurred during the experiments.

2. In Fig. S2 and page 4. In the non-denaturing-PAGE experiments, 150 mM NaCl is included in the reaction, so most G-actin should have been polymerized to F-actin. However, CAP2 can bind to G-actin and prevents polymerization. It should be appropriate to describe simply as "actin" rather than "G-actin" (page 4, last line).

3. It is an interesting observation that CAP2 expression is high in the young stage and gradually declines. Is anything known about changes in the pointed-end actin dynamics during aging or postnatal development? No experiments are necessary, but additional published information, if any,

would enrich discussion.

4. CAP is a conserved protein, and essential roles of CAP in muscle have also been demonstrated in model organisms such as zebrafish and nematodes. Adding these references to Introduction should broaden the significance of this research.

5. It is somewhat surprising that CAP2 does not appear to have effects on thin filament lengths. It should be informative to add localization of Tmod in CAP2-KO. If Tmod localization is not strongly affected by CAP2 KO, it explains why thin filament lengths are not affected.

We would like to thank the Reviewers for their valuable input, which allowed us to improve our study and clarify the representation of our data. The Reviewers had many positive comments including: "...this study adds to our understanding of how CAP2 contributes to sarcomeric actin organization in cardiomyocytes", "These novel findings pave the way for additional research into how actin disassembly factors in cardiomyocytes contribute to cardiomyocyte maturation. These findings are supported by their data, which is clearly presented and well explained", "Each experiment was strictly controlled and well documented, an appropriate statistical analysis was applied. I am impressed with this thorough work, which unveils detailed molecular mechanisms of the thin filaments maturation in cardiac muscle", "This work presents excellent functional characterization of CAP2 in mouse cardiomyocytes..." and "The provided information significantly advances our understanding of the cell biological mechanism of sarcomere assembly". Our point-by-point responses to the Reviewers' comments are listed below.

Reviewer #1:

1) In the section titled: CAP2 associates with a dynamic population of actin filaments the authors treat cells with a low concentration of actin depolymerizing drug Lat A and observe a loss of CAP2 at filament pointed ends. They conclude, "This result suggests that assembly of CAP2 at the pointed end is dependent on active actin polymerization in cardiomyocytes." However, CAP2 also binds to G-actin making it seem possible that the increased free G-actin in the cytosol could compete CAP2 away from pointed ends. Suggested changes shown below:

a. Is there evidence that Lat A disrupts the interaction between actin monomers and CAP2? If so, the authors should indicate this and provide the citation.

To our knowledge, the effect of Lat A on the interaction between actin monomers and CAP2 has not been demonstrated until this study.

b. If it is possible that increased monomeric actin is competing CAP2 away from actin pointed ends, this alternative interpretation should be added to their interpretation of the results in this section.

We agree. We have now added the following statement: "Alternatively, an increase in G-actin by Lat A treatment recruits CAP2, but not Tmod1, away from pointed ends." P4, lines 127-128.

2) Figure S1 is titled: Excess levels of CAP2 has an insignificant effect on thin filament lengths. This is unclear because the differences appear statistically significant.

a. Suggested change: Change the language to "minor effect" or another alternative phrase.

The legend has been changed to "Excess levels of CAP2 have a minor effect on thin filament lengths." P3, line 24 in Supplementary Information.

3) In Figure S1, filament lengths are shown in a bar graph that does not show individual data points. It would also make the data easier to interpret if the data were represented as a dot plot so that each datapoint is visible. For example, is there a population of filaments that have shorter lengths?

a. Suggested change: Replace the bar graph with a graph that shows individual measurements.

The bar graphs for the thin filament and sarcomere lengths in Figure S1 have been changed to dot plots.

4) In the sections CAP2 depolymerizes actin filaments and Tropomyosin inhibits the depolymerization activity of CAP2 the authors describe changes in the percentage of F-actin and G-actin without providing the actual percentage measured or the fold change or the under the conditions investigated. This makes it difficult to follow the magnitude of the change in different experiments.

The mean percentages and standard deviations of treatment groups in Figure 4b, d and h have been

added to the Results and Figure Legend. P4, lines 148-149, 152-153; P5, lines 188, 191, 198-200; P22, lines, 975-976, 980-981; P23, lines 997-999, 1003-1004, 1006.

5) In Figure 4E there is a s15 kDa size marker that needs to be moved to the left. The alignment of the marker has been corrected.

6) In Figure 4F, the label on the vertical axis should be changed to “Band density of F-actin in the pellet” to clarify what is being quantified on the gel.

The label has been changed. Please see updated figure above (#5).

7) In Figure 4 the authors conclude that “CAP2 depolymerizes actin filaments but stabilization by tropomyosin prevents CAP2-mediated filament disassembly.” This is supported by the blot and accompanying graph in panels 4E and 4F. However, given that tropomyosin has been previously shown to stabilize actin filaments and protect them from disassembly by cofilin, this finding is not unexpected. The more interesting, and potentially relevant, finding is that CAP2 appears to help cofilin overcome the stabilization of F-actin by tropomyosin. This may be particularly relevant given that thin filaments in sarcomeres are decorated with tropomyosin. Suggested changes below:

a. In their assay, is there a concentration of CAP2 that when added to cofilin allows cofilin to disassemble tropomyosin decorated filaments to the same degree seen with undecorated filaments? This can be addressed in the text associated with figure 4 in section Tropomyosin inhibits the depolymerization activity of CAP2.

At all tested CAP2 concentrations, actin disassembly was stronger compared to cofilin (CFL2) only (control). This function was inhibited by the presence of tropomyosin (Tpm) at all CAP2 concentrations tested. Therefore, within our experimental setup there was not a concentration in which similar magnitudes of actin disassembly by CFL2 and CAP2 was observed in the absence or presence of Tpm. We have now added the following statement: “At all concentrations of CAP2 tested, filament disassembly was stronger than with CFL2 only. Furthermore, the presence of Tpm inhibited CAP2’s depolymerization activity at all CAP2 concentrations tested.” P5, lines 180-183.

b. In the graph in figure 4F it appears that datapoints are normalized to the amount of actin in the pellet observed when cofilin is added. This needs to be clarified in the figure caption. Are the trends the same if all data points are normalized to the actin only condition?

The definition of the control (actin in the presence of CFL2) has been added to the legend. P23, line 987. The trend was the same when all data points were normalized to the actin only condition; the comparison is shown in the figures below.

8) In the section titled: Tpm1.1 recruits Tmod1, not CAP2 to thin filaments, the authors conclude “Therefore, Tpm1.1 favors the recruitment of Tmod1 (due to its known tropomyosin-binding sites) but not CAP2, to thin filaments, suggesting that CAP2 cannot directly bind to filaments stabilized by Tpm1.1.” However, it is not clear why they come to this conclusion given that the fraction of CAP2 that pellets with F-actin is unchanged with increased Tpm1.1 expression (see Fig. 4J).

a. Suggested change: The authors should clarify how their data indicates CAP2 is incapable of interacting with tropomyosin decorated filaments or else change the wording in this section.

We observed that overexpression of Tpm increases the amount of F-actin recovered from cardiomyocytes (Fig. 4g-h). We predict that this is caused by stabilization of more dynamic filaments. An increase in the amount of Tmod recovered in the pellets indicates that the pointed ends of these filaments were decorated by Tmod. A change in the levels of CAP2 was not observed, suggesting that these new stabilized filaments do not recruit CAP2, likely as a result of their saturation by Tpm. The description of these results has now been expanded: “Therefore, the additional filaments that are stabilized by excess levels of Tpm1.1 (Fig. 4h) recruit Tmod1 (due to its known tropomyosin-binding sites and thus increased affinity) but not CAP2, to their pointed ends. This suggests that the binding of CAP2 to actin filaments is ineffective when stabilized by Tpm1.1.” P5-6, lines 201-205.

9) In figure 4B, the authors show that increased CAP2 expression shifts more actin into the monomer fraction and decreases F-actin. This seems inconsistent with the finding the TFL does not change.

a. Suggested change: The authors should add additional text to the discussion to address how it is that there is less actin in polymer form in CAP2 overexpressing cells but the overall TFL does not change (Fig. S1).

We added the following text to further explain these findings: “Although varying the amount of CAP2 does not alter TFLs, we found it to affect the levels of G- and F-actin in cells. Based on this observation, we predict that unlike Tmod1 and Lmod2, CAP2 does not “fine-tune” the lengths of existing filaments but it plays a role in the disassembly of entire filaments.” P10, lines 407-410.

0) In the section titled: CAP2 inhibits incorporation of actin onto thin filament pointed ends. The authors show via FRAP that actin recovery at the pointed ends is diminished with CAP2 overexpression but the mobile fraction is unchanged. They state, “Although, the rate of actin incorporation at the pointed ends was diminished by CAP2, the mobile fraction at the slow phase of recovery was unaffected, indicating that the amount of actin assembled at the end of the experiment was unchanged between cells transduced with mCherry-CAP2 and mCherry.” This is in agreement with their finding that TFL does not change with increased expression of CAP2 (Fig. S1). However it seems to disagree with their finding in figure 4B where they show that CAP2 overexpression results in a 10% increase in G-actin. The authors should make changes to the text to answer the questions below:

a. What is the difference between what is being measured in their pelleting assay and FRAP experiments such that the pelleting assay indicates CAP2 reduces monomer incorporation into polymer while FRAP does not?

The bulk pelleting assay measures total actin filaments irregardless of orientation, whereas thin filament length and FRAP measurements are collected from groups of aligned single filaments. Due to these differences, obtaining different outcomes are not unexpected. We believe that the increase in G-actin levels in the presence of CAP2 in pelleting assays result from total disassembly of a population of dynamic filaments that are likely not stabilized by Tpm and/or Tmod1. Due to their dynamic nature, these filaments were not “captured” and investigated in thin filament length and FRAP experiments. This important clarification has now been described above (#9).

b. How are the authors accounting for recovery due to GFP labeled G-actin diffusing into the bleached area from the cytosol without being incorporated into F-actin?

At certain instances the fluorescence recovery plots were “noisy”, which prevented us from fitting a two-phase exponential equation to the data and extracting reliable fit parameters ($R^2 < 0.70$). This could be

reasoned by GFP-labeled G-actin diffusing into the bleached area, as the reviewer remarked. These measurements were excluded from analysis as stated in our “Reporting Summary”. We have also updated the Methods: “If fitting the two-phase exponential equation to the fluorescence recovery data did not provide an R^2 value greater than 0.70 or did not converge, the calculated parameters were deemed unreliable and not included in analysis.” P17, lines 710-713.

c. If the concentration of free G-actin in the cytosol is different between their two different conditions (as they indicate in figure 4B) could this influence their measurements?

Different levels of monomeric GFP-cActin could potentially contribute to our results. However, one would expect excess cytosolic actin to incorporate to both ends of filaments. Since we observed a difference only at the pointed ends when mCherry-CAP2 was expressed, we believe that the presence of free G-actin most likely has a minor effect in the fluorescence recovery experiments.

11) In figure 6 discussed in the section titled, "Cap2-null cardiomyocytes display altered α -actin assembly and expression", the authors show single representative images of different actin isoform staining for the WT and KO cells.

a. Suggested change: The authors should indicate in the text or the figure how repeatable these results are. How many cells were looked at? Did all cells display the same phenotype (disorganized sarcomeres, reduced staining, etc.)?

The following text has now been added to the Methods: “Fluorescence imaging was conducted with cells from at least three independent cultures or animals with 15-30 cells investigated per culture or animal. Representative images were selected based on the staining pattern observed in the majority of cells.” P15, lines 622-624.

12) In the section titled: Cap2-null cardiomyocytes display altered α -actin assembly and expression, the authors state “Although cActin assembled into thin filaments in WT and Cap2-KO cells, its pattern was more irregular in Cap2-KO cardiomyocytes (Fig. 6D).” It is unclear what irregular means. Do the authors mean that there is higher cell-to-cell variability or variability within a single cell?

a. Suggested change: Clarify the description of the phenotype in the text.

We agree with the Reviewer that our description as “irregular” is too vague. We added the following sentence: “Although cActin assembled into thin filaments in WT and Cap2-KO cells, it demonstrated an irregular alignment with the phalloidin stain, such that cActin striations were disorganized in Cap2-KO cardiomyocytes (Fig. 6d).” P7, lines 280-283.

13) In the example image shown in figure 6D, it appears that there is less cActin in the KO cells. Additionally, in the quantification of cActin by immunoblot in figure 6F, in two out of three cultures it appears there is less cActin in KO cells than in WT cells.

a. Suggested change: If the example image gives a false impression of the levels of cActin in KO cells, a different image should be selected.

We changed the representative image for the blot in Fig. 6f to better reflect the data.

b. Suggested change: If in some cultures of the KO line there is less cActin, this should be discussed in the text to indicate why there is culture-to-culture variability in the amount of cActin measured in KO cells. This may also be clarified by repeating the experiment to determine if one of the measurements is an outlier and that there is actually a trend to have less cActin in KO cells.

Please see our response to comment #13a.

14) In the section titled: Cap2-null cardiomyocytes display altered α -actin assembly and expression, the authors state “Collectively, our results indicate that expression levels and assembly of α -SMA and α -SKA are significantly altered in Cap2- KO cardiomyocytes, indicating CAP2 plays an important role in the distribution of α -actin isoforms during cardiac muscle development.” It is unclear what "distribution" means in this sentence.

a. Suggested change: The authors should change the text to clarify how CAP2 regulates the distribution of alpha-actin isoforms. Distribution between F-actin and G-actin? Distribution within a single filament?

This statement has now been clarified: “Collectively, our results indicate that expression levels and assembly of α -SMA and α -SKA are significantly altered in *Cap2*-KO cardiomyocytes, indicating CAP2 plays an important role in the distribution of α -actin isoforms between G- and F-actin during cardiac muscle development.” P7, lines 289-292.

15) "In the section α -SMA and α -SKA replace cActin in thin filaments from Cap2-KO mice", the authors state “Third, immunostaining of Cap2-KO hearts revealed that striations of cActin were narrower compared to WT hearts (Fig. 7D).” This is an interesting observation given that TFL does not change in Cap2-KO mice. The authors should expand on what this observation means by addressing the questions below:

a. Does this observation indicate that loss of CAP2 restricts cActin to a specific location within the thin filament? For example, is cActin enriched at the barbed or pointed ends of thin filaments in KO cells?

Based on a co-stain with phalloidin, we found cActin to be lost from the pointed ends of *Cap2*-KO filaments. We now include an image of this observation in Fig. 7g and also added text to the Results: “Third, immunostaining of thin filaments from Cap2-KO hearts revealed that striations of cActin were shorter (from their pointed ends) compared to WT hearts (Fig. 7d, g).” P8, lines 309-311.

b. Are the other two actin isoforms uniformly distributed within thin filaments?

Due to very high expression (~10~30-fold over WT) of α -SMA and α -SKA in *Cap2*-KO hearts (Fig. 7c) and their weak staining pattern in WT mice, we were unable to make a similar comparison for these isoforms as we could for cActin. However, since we observed unchanged total α -actin levels (Fig. 7a-c) and TFLs (Fig. S9) between the WT and KO animals, we predict that α -SMA and α -SKA replace cActin from filament pointed ends.

16) In the discussion section the authors state “An increase in actin dynamics at the barbed ends of Cap2-KO cardiomyocytes may be explained by interactions between CAP2 and G-actin.” However, CAP has also been shown to inhibit the activity of the formin INF2 (DOI: 10.1038/s41556-019-0307-4 and DOI: 10.1073/pnas.1914072117). Additionally, INF2 is expressed in cardiomyocytes and localizes to Z-bands (DOI: 10.1091/mbc.E13-08-0443).

We have added this reference and included the following statement in the Discussion: “An alternative explanation for increased recovery of actin at the barbed ends of *Cap2*-KO cells could be due to CAP2’s inhibitory role (that was discovered in *vitro*) on the actin polymerization activity of inverted formin-2 (INF2), a barbed-end interacting protein⁵⁵.” P10, lines 391-394.

17) The authors discuss the role of CAP2 in stabilizing actin giving contradictory conclusions in different sections. In the discussion section the authors describe their FRAP results stating, “Excess levels of CAP2 strongly stabilized the highly dynamic population of actin only at the pointed ends by decreasing the rate constant of recovery.” However, in the section titled CAP2 depolymerizes actin filaments the authors state, “To decipher CAP2’s specific function at thin filament pointed ends, we tested the effect of CAP2 on the stabilization of actin filaments... Collectively, this result is the first to hint at CAP2’s distinct role in cardiac muscle: it promotes depolymerization of F-actin into G-actin and it is a new regulator of actin filament disassembly in cardiomyocytes.”

a. Suggested change: The language in the manuscript, and specifically use of the word ‘stabilize’ should be made consistent. When the authors refer to actin stabilization do they mean decreasing loss of actin monomer from F-actin or decreasing the addition of new monomers or both?

We agree with the Reviewer that “stabilized” is not applicable to describe CAP2’s role in our FRAP experiments. We made the following changes in the Results: “Excess levels of CAP2 strongly decreased the rate of a highly dynamic population of actin only at the pointed ends.” P9, lines 351-353.

Reviewer #2

1) In the experiment illustrated in Fig. 4 E-F the Authors showed that Tpm1.1 inhibited the acceleration of CFL2-induced actin depolymerization by CAP2. However, at the concentrations of cofilin used in this experiment there was practically no tropomyosin bound. How could tropomyosin protect actin from depolymerization if it was removed by cofilin? Also, in Fig. 4E the CFL2 +/- row should be aligned with the appropriate gel bands.

To quantify the Tpm replacement by CFL2, we compared the band density ratios between Tpm and actin in the absence and presence of CFL2 from the experimental data presented in Fig. 4e. The graph below shows that in the presence of CFL2, ~50% of the remaining filaments were still coated with Tpm. Therefore, we respectfully disagree with the Reviewer that there was no tropomyosin bound under these conditions. Nevertheless, in line with the Reviewer’s concern, the data below indicate that in the presence of CFL2 the filaments are not fully saturated with Tpm and areas of “bare” actin remain. One could argue that either the barbed or pointed end of the filaments favor displacement of Tpm by CFL2 and studying this could shed light onto the biophysical properties of these proteins upon interaction. However, our co-sedimentation assay does not have the resolution to extract this information, thus, we believe that it is beyond the scope of our study to address this issue in our manuscript.

The row in Fig. 4E has been properly aligned. See Reviewer 1, comment #5 for the updated figure.

2) I cannot agree with the Authors that tropomyosin reduces the binding of cofilin. In the Discussion section (page 10, last paragraph) they refer to some older work, which suggested such effects, but the recent studies on muscle tropomyosin isoforms did not confirm such mode of Cof-2 action (e.g. doi:10.3390/ijms21124285).

Therefore, the hypothesis that CAP2 together with CFL2 is responsible for depolymerization of the population of actin filaments that are not stabilized by tropomyosin should be revised.

We thank the Reviewer for informing us about this recent study (doi:10.3390/ijms21124285), which shows that the affinity of CFL2 for F-actin is similar in the absence or presence of Tpm1.1. Interestingly, Robaszkiewicz et al. also demonstrated other findings in which Tpm1.1 was found to inhibit CFL2's ability to depolymerize actin, similar to our observation in this manuscript. We have now revised our discussion to replace the older work with Robaszkiewicz et al. and removed the parts that mention an inhibition of CFL2's binding to F-actin by Tpm1.1. We now state: "Tpm1.1 inhibits the depolymerization activity of CFL2⁶¹ and also CAP2 (shown above)." P11, lines 457-459.

3) Using various methods the Authors showed that CAP2 tends to associate with a more dynamic population of thin filaments. I must admit that it is not entirely clear to me what the population of dynamic filaments in cardiomyocytes is. Fig. 1C shows a perfect alignment of CAP2 and Tmod1. Does it mean that in sarcomere the I-bands contain a mixture of parallel thin filaments that are either stabilized by Tpm-Tmod or are Tpm-free, but capped by CAP2?

For clarification, we made the following change: "These results reveal that Tmod1 associates with more mature and stabilized filaments, whereas CAP2 associates with a dynamic population of parallel filaments that are potentially not stabilized by Tmod1 and/or Tpm." P4, lines 124-126.

4) Although CAP2 is the main focus of these studies, in the Introduction the Authors have not included any information on CAP2 structure and modes of actin binding. A short description of the multidomain structure and oligomerization ability would give readers a better background and would help to understand the complex functions of CAP2 in cardiomyocytes.

The following text has been added to the Introduction: "The N-terminal region of CAP consists of a coiled-coil and a helical-folded domain (HFD)^{30,31}, whereas the C-terminal region contains two poly-proline rich domains, a Wiskott-Aldrich-homology 2 (WH2) domain and a CAP-retinitis pigmentosa (CARP) domain that is comprised of β -sheets^{32,33}. The coiled-coil region of human CAP1 and CAP2 allows tetramers of HFDs to form, which increases cofilin-dependent actin depolymerization³⁴. Mammalian CAPs interact with filamentous actin (F-actin) via the HFD^{31,34,35} and with globular actin (G-actin) using the WH2 and CARP domains³³." P2-3, lines 62-69.

5) Two recent papers have proposed models of CAP-driven dynamics of actin filaments (doi: 10.1038/s41467-019-13268-1 and doi: 10.1038/s41467-019-13213-2). In these papers yeast isoform Svr2/CAP and CAP1 were used. In the light of the data collected in the present study, can these models be applied to CAP2 function in cardiomyocytes?

We cited these studies in the Introduction (P3-4, lines 63, 68) and Discussion (P11, line 447). These models can potentially be applied to CAP2 function in cardiomyocytes.

Reviewer #3

1) In Fig. 4E, the protective role of tropomyosin for CAP2 and CFL2 is shown by pelleting assays. The gels are only shown for the pellet fractions. However, gel images for the supernatant fractions should also be included. This is important to show the equal amounts of total actin in all the reactions and no protein degradation occurred during the experiments.

Fig. 4e now includes the gel images for the supernatant samples and quantification of actin band densities. Please see Reviewer 1, comment #5 for the updated figure.

2) In Fig. S2 and page 4. In the non-denaturing-PAGE experiments, 150 mM NaCl is included in the reaction, so most G-actin should have been polymerized to F-actin. However, CAP2 can bind to G-actin and prevents polymerization. It should be appropriate to describe simply as "actin" rather than "G-actin" (page 4, last line).

We changed "G-actin" to "actin" in the image and legend (P4, line 48 in Supplementary Information) and Methods (P13, line 536).

3) It is an interesting observation that CAP2 expression is high in the young stage and gradually declines. Is anything known about changes in the pointed-end actin dynamics during aging or postnatal development? No experiments are necessary, but additional published information, if any, would enrich discussion.

To our knowledge, this information does not exist in the literature.

4) CAP is a conserved protein, and essential roles of CAP in muscle have also been demonstrated in model organisms such as zebrafish and nematodes. Adding these references to Introduction should broaden the significance of this research.

Two references about CAP's role in zebrafish (doi.org/10.1016/j.yexcr.2012.09.013) and nematodes (doi: 10.1242/jcs.104950) have been added. P2, line 58.

5) It is somewhat surprising that CAP2 does not appear to have effects on thin filament lengths. It should be informative to add localization of Tmod in CAP2-KO. If Tmod localization is not strongly affected by CAP2 KO, it explains why thin filament lengths are not affected.

We agree. Tmod localization does not change noticeably in *Cap2*-KO cells. We now expanded Fig. S6 to include immunofluorescence images of Tmod1 localization. We also updated the text: "TFLs and Tmod1 localization in *Cap2*-null cardiomyocytes were comparable to WT cardiomyocytes (Fig. S6), further highlighting that CAP2 has little effect on determining TFLs." P7, lines 264-266.

REVIEWERS' COMMENTS:

Reviewer #1 (Remarks to the Author):

The authors have satisfactorily addressed all of my comments. Where necessary, they added additional information to the text and updated figures. These revisions included adding additional references to the literature, changing text in the results section to clarify their interpretations of the data, expanding the description of their experiments in the methods section, and updating figures and figure legends to better represent the repeatability of the observed phenotypes. These changes clarify the authors' findings and strengthen their conclusions about (1) the effect of tropomyosin in stabilizing actin filaments against enhanced cofilin disassembly in the presence of CAP2, (2) the role of CAP2 in slowing the assembly of actin monomers into thin filaments in cardiomyocytes, and (3) how CAP2 regulates the expression of actin isoforms and their assembly into (and distribution within) thin filaments during cardiomyocyte development. I recommend the work for publication.

Reviewer #2 (Remarks to the Author):

All my comments and concerns were appropriately addressed in the rebuttal letter and the revised manuscript. Changes that have been made improved the manuscript.

Reviewer #3 (Remarks to the Author):

The revision clarified all previous concerns.